# In situ 3D bioprinting with bioconcrete bioink

Mingjun Xie [1,2,8], Yang Shi[3,4,5,8], Chun Zhang [3,4,5], Mingjie Ge[3,4,5], Jingbo Zhang[1,2], Zichen Chen[1,2], Jianzhong Fu[1,2], Zhijian Xie [3,4,5] & Yong He [1,2,6,7 ✉]

In-situ bioprinting is attractive for directly depositing the therapy bioink at the defective organs to repair them, especially for occupations such as soldiers, athletes, and drivers who can be injured in emergency. However, traditional bioink displays obvious limitations in its complex operation environments. Here, we design a bioconcrete bioink with electrosprayed cell-laden microgels as the aggregate and gelatin methacryloyl (GelMA) precursor solution as the cement. Promising printability is guaranteed with a wide temperature range benefiting from robust rheological properties of photocrosslinked microgel aggregate and fluidity of GelMA cement. Composite components simultaneously self-adapt to biocompatibility and different tissue mechanical microenvironment. Strong binding on tissue-hydrogel interface is achieved by hydrogen bonds and friction when the cement is photocrosslinked. This bioink owns good portability and can be easily prepared in urgent accidents. Meanwhile, microgels can be cultured to mini tissues and then mixed as bioink aggregates, indicating our bioconcrete can be functionalized faster than normal bioinks. The cranial defects repair results verify the superiority of this bioink and its potential in clinical settings required in in-situ treatment.

[1] State Key Laboratory of Fluid Power and Mechatronic Systems, School of Mechanical Engineering, Zhejiang University, 310027 Hangzhou, China. [2] Key Laboratory of 3D Printing Process and Equipment of Zhejiang Province, School of Mechanical Engineering, Zhejiang University, 310027 Hangzhou, China. [3] Stomatology Hospital, School of Stomatology, Zhejiang University School of Medicine, 310006 Hangzhou, China. [4] Zhejiang Provincial Clinical Research Center for Oral Diseases, 310006 Hangzhou, China. [5] Key Laboratory of Oral Biomedical Research of Zhejiang Province, 310006 Hangzhou, China. [6] Cancer Center, Zhejiang University, 310058 Hangzhou, Zhejiang, China. [7] Key Laboratory of Materials Processing and Mold, Zhengzhou University, 450002 Zhengzhou, China. [8] These authors contributed equally: Mingjun Xie, Yang Shi. ✉email: yongqin@zju.edu.cn

As an emerging organ defect treatment, "in-situ bioprinting"[1] initially proposed by Campbell[2]. has been capturing attentions in the clinic. In brief, therapy bioink is directly deposited on the patients' wounds by surgical bioprinters along paths according to their 3D morphologies[3]. Currently, it mainly utilizes similar methods for in-vitro bioprinting and has been applied in skin, cartilage, bone treatments[4]. Compared to organ implantation based on in-vitro 3D bioprinting, it has more advantages for its in-situ deposition feature (Supplementary Note 1).

However, in-situ bioprinting is rudimentary and has been restricted in clinical applications. Besides the lack of reliable in-situ bioprinters[4], one of the main reasons is that there is less suitable bioink meeting its special requirements. In existing relevant studies, the applied bioink is mostly similar to the one in in-vitro bioprinting, namely precursor solution, which is not a promising choice for in-situ bioprinting. (i) In most in-situ bioprinting cases, there are no conditions to strictly control the rheological properties of bioink, especially thermo-sensitive bioink. (ii) Unlike the receiving basements with clean surface and controllable temperature on in-vitro bioprinters, in-situ bioprinting has a special receiving basement, namely the patient's wounds with a constant temperature (37 °C) and blood, which can collapse the printed structure before crosslinking. (iii) The crosslinked bioink should own low mechanical modulus for encapsulated cells to exert therapy functions. (iv) Structures should have high mechanical properties matching the defect, protecting itself from damage during repair, which, however, leads to a huge contradiction to requirement (iii). Building composite structures, that is, printing strong scaffolds followed by casting soft hydrogel, has become an effective solution[5–8]. However, such a complex printing process cannot be realized in in-situ bioprinting. (v) Strong binding force should be formed on defect-printed structure interface. (vi) In-situ bioink should be portable and easily prepared for occupations such as soldiers, athletes, and drivers who can be injured in emergency.

Microgels have become popular bioprinting structures in cell therapy[9], controlled drug release[10], disease modeling[11], etc., and many fabrication methods have been proposed[12–16]. Recently, besides independent function unit, in the review on microgels published in Nature Reviews by Burdick et al. [17] in 2020, the wide application of the "secondary bioprinting"[18–21] of microgels as a bioink component in the future has been predicted. In latest work of Alge et al. [22] published in Science Advances in 2021, an in-depth investigation on the microgels dissipation process during printing was presented. Wang et al. [9] injected alginate microgels to repair rat organ defect. Burdick et al. [23,24] extruded gathered microgels to establish specific 3D structures. All of the research benefited from not only the promising biocompatibility of microgels but also their unique rheological properties similar to Bingham fluid[25–27], which displays as elastomer below certain stress but flows as Newton fluid once the stress was further increased. Therefore, microgel-based bioink have the potential to be further designed as a bran-new clinical in-situ bioprinting bioink to adapt the complicated requirements.

In this work, we develop the bioconcrete bioink (A–C bioink) for in-situ bioprinting, the name of which comes from concrete for construction and its abbreviation comes from the two main components: aggregate (A) and cement (C) (Fig. 1, Supplementary Videos 1 and and 2). Electrosprayed GelMA microgels (500 μm) are used as the main (A) component to obtain robust rheological properties similar to Bingham fluid in complex environments (Supplementary Notes 3 and 4). GelMA precursor solution (with photoinitiator) are used as the auxiliary (C) component to ensure the fluidity and printability. The photo-crosslinked composite structure owns A/C structure with low/

high mechanical modulus, respectively, which perfectly solves the contradiction between maintaining biocompatibility for loaded therapy cells and bearing the high tensile/compressive stress around the defect. Additionally, C component can easily infiltrate the wound surface and form high internal friction and hydrogen bonds on defect–hydrogel interface after photocrosslinking. Conveniently, this bioink is portable because A/C component can be conserved in liquid nitrogen, which can be thawed with heating devices or body temperature in accidents. Meanwhile, the microgels can be cultured to mini tissues before mixing, indicating our bioconcrete bioink can be functionalized faster than traditional ones. The in-situ treatment results of rat cranial defects verify its potential in clinical settings in in-situ bioprinting in the future.

## Results

**Volume proportions of A/C component**. The degree of substitution (DS) of GelMA can be easily modified by changing the ratios of gelatin and methacrylic acid. Higher DS and precursor solution concentration result in higher mechanical properties of the photocrosslinked structures[28,29]. According to our experience, 5% (w/v) low-DS GelMA (EFL-GM-30) has low mechanical properties and high biocompatibility. Thus, it was selected to electrospray GelMA microgels (A30/5, 500 μm), which has been verified to own very uniform diameters (Supplementary Note 4 and Supplementary Fig. 6). 20% (w/v) high-DS GelMA (EFL-GM-300) can stand compressive stress, such as bone tissue, whereas 20% (w/v) low-DS GelMA (EFL-GM-30) can stand stretching stress, such as tendon tissue. Thus, they were selected as C component (C300/20, C30/20). For comparison, 5% (w/v) low-DS GelMA (EFL-GM-30) was also selected as C component (C30/5).

We assumed that C component exactly infiltrated the vacancy among A component so that the microgels could generate enough internal friction to avoid collapse before crosslinking. Accumulated A30/5 (with GFP) in the cell sieve were repeatedly soaked in C30/5 (with RFP) and lifted to filter the redundant C30/5 (Fig. 2a). C30/5 spontaneously entered the vacancy among A30/5 with capillary force. The crosslinked A–C bioink showed that A component closely attached to each other (Fig. 2b, c). The volume proportion of A component was analyzed red/green areas as 73.215 ± 2.312% (Supplementary Note 2), which was similar to the atomic space utilization in hexagonal closest packed (HCP) in crystal chemistry (74.05%) (Fig. 2d) in that A component displayed standard spheroid shape under floatage and surface tension in liquid C component and the system tended to own the lowest energy under gravity.

**Emergency bag for portability**. An emergency bag was designed to meet the storage and portability requirements (Fig. 2e). A component loading cells was immersed in cryopreservation medium and stored in a freezing tube as previously described by Ouyang et al. [30]. C component was also stored in another freezing tube. The volume proportions of A/C component in one kit were 74.05%/25.95%, respectively. Both freezing tubes were stored in a container filled with liquid nitrogen.

In an accident, the freezing tubes were removed from the container and thawed with a tiny heating pad at 37 °C powered by a portable USB battery (Fig. 2f). For very urgent accidents without heating devices, they could be directly heated using the body temperature. Certainly, the cold injury on the patient's skin should be noticed. Then, the cryopreservation medium was removed by a syringe and a napkin. A component was transferred to C component with a thin spoon and stirred uniformly, followed by transfer A–C bioink into a new syringe with cone

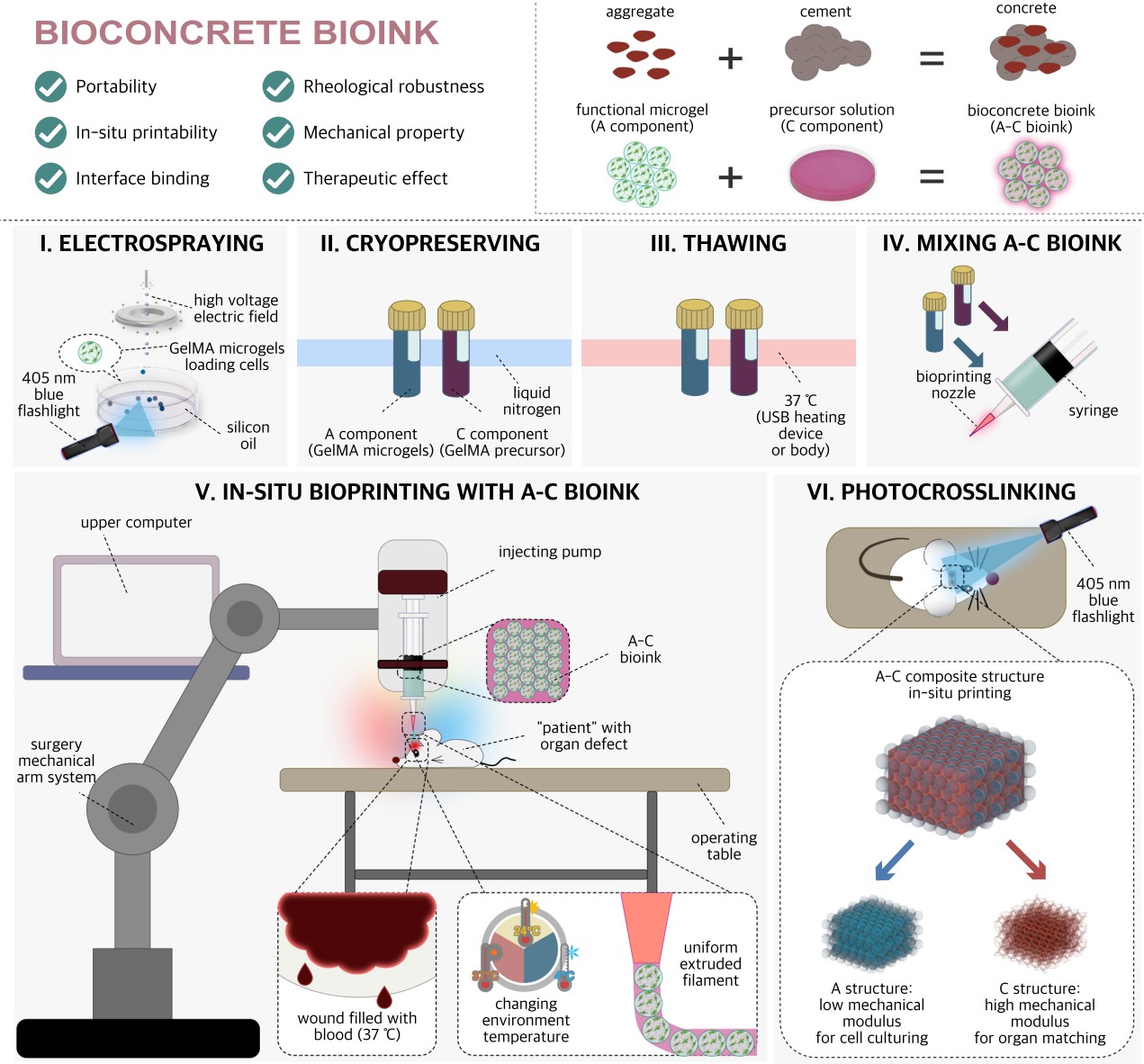

**Fig. 1 Train of thought of A–C bioink designing and sketch of the preparing/using method.** Upper panels show the properties of the proposed A–C bioink inspired by concrete for construction. Lower panels (I-VI) show the usage of A–C bioink.

printing nozzle for in-situ bioprinting with professional in-situ bioprinters or just hands, followed by photocrosslinking with 405-nm blue flashlight, which has been verified to be safe and widely used in current clinical scenes, such as tooth photocuring, blue light treatment of neonatal jaundice, etc.

In practice, the total volume of bioink is determined by the volumes and quantities of the applied freezing tubes. At A–C bioink production end, the prepared A–C bioink can be loaded with different types of freezing tubes according to the production requirements. Moreover, at the rescuing end, in terms of the quantities of freezing tubes, rescuers can definitely take as many freezing tubes loading A–C bioink as possible according to the patient injury situation. Therefore, with the further development and mass production of A–C bioink, this bioink system would be feasible for the in-situ bioprinting of tissue defect as large as possible.

**Rheological robustness and mechanism.** A–C bioink should own high rheological robustness in different temperature

conditions to adapt to different accident situations of in-situ treatment. A30/5–C30/20, A30/5–C300/20, A30/5–C30/5 and C bioink C30/20, C300/20, C30/5 were prepared. 4, 24 and 37 °C were selected, under which GelMA precursor solution would be in excessive gelation, optimized sol–gel and excessive solization state for extruding bioprinting, respectively[31,32].

Flow sweep results indicated both A–C bioinks and C bioinks had shear-thinning feature under the three selected temperatures (Fig. 3c), meeting the basic requirement of extruding bioprinting. This was because GelMA microgels were separated from each other and the friction among them disappeared at a high shear rate. Furthermore, the orientation of discrete GelMA molecules in C component tended to be coincident and molecule twining was reduced (Fig. 3a). According to published studies[9,33], the system mainly composed of microgels had Bingham fluid properties. The flow sweep data of A–C bioinks were further fitted with the Bingham fluid model as follows:

$$\sigma = \sigma_y + \eta_B \dot{\gamma} \qquad (1)$$

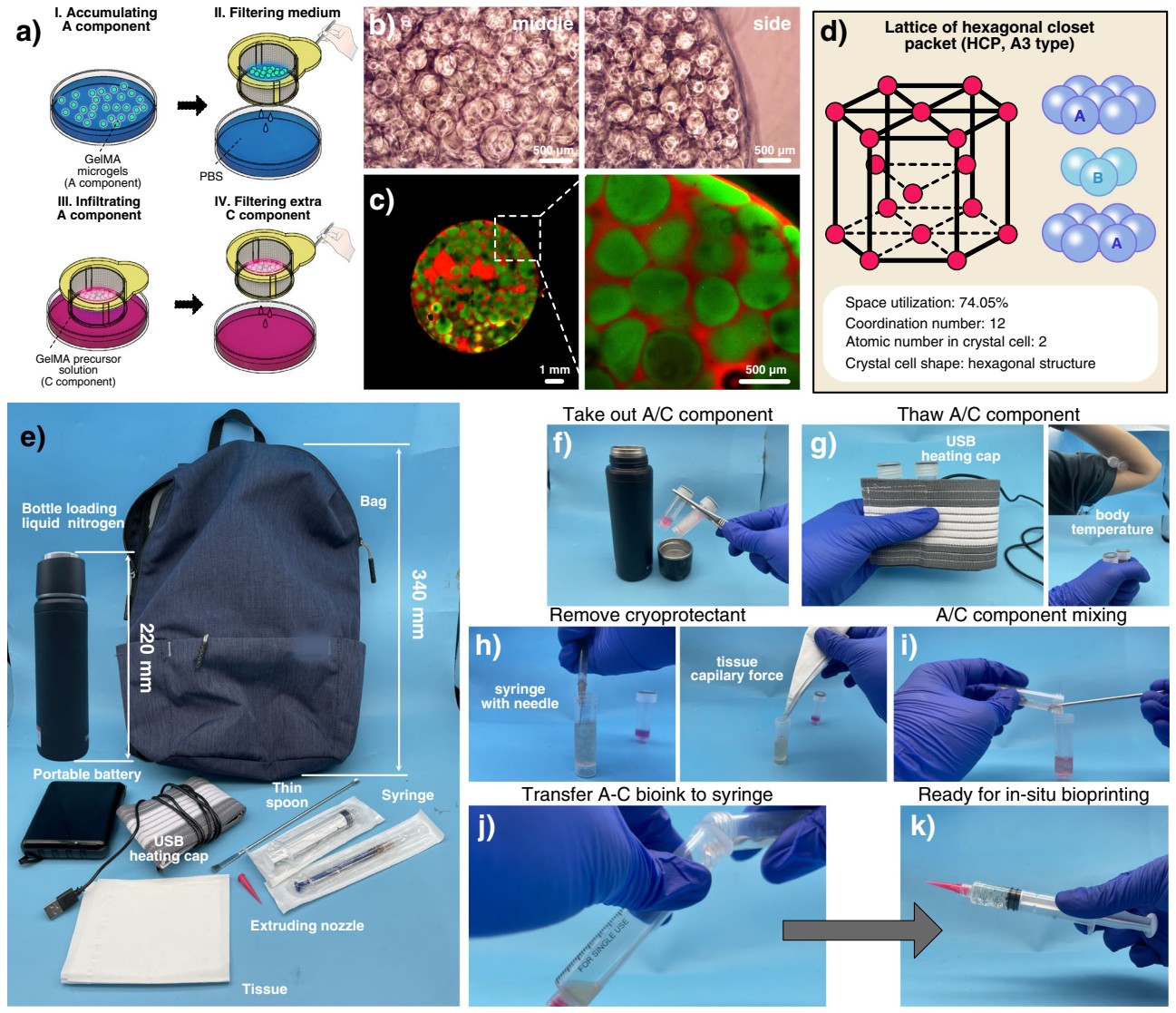

**Fig. 2 Volume proportion and portability of A-C bioink. a** Sketch of the wetting method in volume proportion analysis experiment. **b** Optical morphology of A–C composite structure. **c** Fluorescent morphology of A–C composite structure (Zen, Carl Zeiss, version 8,0,0,273). **d** Lattice of hexagonal closet packet in crystal chemistry. **e** The parts of A–C bioink kit. **f**-**k** The preparation steps of portable A–C bioink. For **b**, **c**, each experiment was repeated independently with similar results for 3 times.

in which $\sigma$, $\sigma_y$, $\eta_B$ and $\dot{\gamma}$ is stress, yield stress, Bingham viscosity, shear rate, respectively. All A–C bioinks showed Bingham fluid feature and linear relationship between stress and shear rate above certain stress (Fig. 3d, e). The yield stress increased at 4 °C due to the excessive gelation of C component during which GelMA molecules would form collagen-like spirochete, spirochete aggregate and aggregate network in succession (Fig. 3b). The solid/fluid feature below/above yield stress was due to the internal friction among A component the malposition of microgels beyond the static friction, respectively.

Bioink should own fluidity to form filaments from nozzles and display solid feature to maintain the shape. The oscillation frequency testing results showed that all A–C bioinks under different temperatures owned a solid/fluid feature under low/high frequency, respectively, benefiting by Bingham fluid property (Fig. 3f). Remarkably, traditional mono-component GelMA bioink would transfer to solization and cannot maintain shape at body temperature (37 °C). However, even under 37 °C, A–C bioink still showed solid state under low frequency, verifying its feasibility for depositing on wounds. The thixotropy results by

adding periodically varied oscillation amplitudes (200% and 1%) indicating the state transfer of A–C bioink was rapid and obvious, confirming its rapid self-healing speed. (Fig. 3g)

Thermo-sensitive bioinks need time to achieve stable state under certain temperature. Besides, for a certain temperature, the sol–gel state at certain time would be affected by the previous state and totally different during the temperature-increasing/decreasing process. Flow temperature ramp testing of A–C and C bioink was carried out to examine the viscosity variation in repeatedly increasing/decreasing temperatures (4–39 °C, 5 °C/min) for three times (I–III) (Fig. 3h). The results showed that even the viscosity curves of the temperature-increasing/decreasing process of the same bioink were not superimposed, verifying poor state reversibility and stability of thermo-sensitive bioink. However, compared to C bioink, the zones surrounded by the viscosity curves in the continuous three tests of A–C bioink were nearly superimposed, indicating A–C bioink could effectively avoid the effect of the previous state. Furthermore, for such a wide temperature changing range, the viscosity of C bioink stretched 4–5 magnitudes, whereas that of A–C bioink

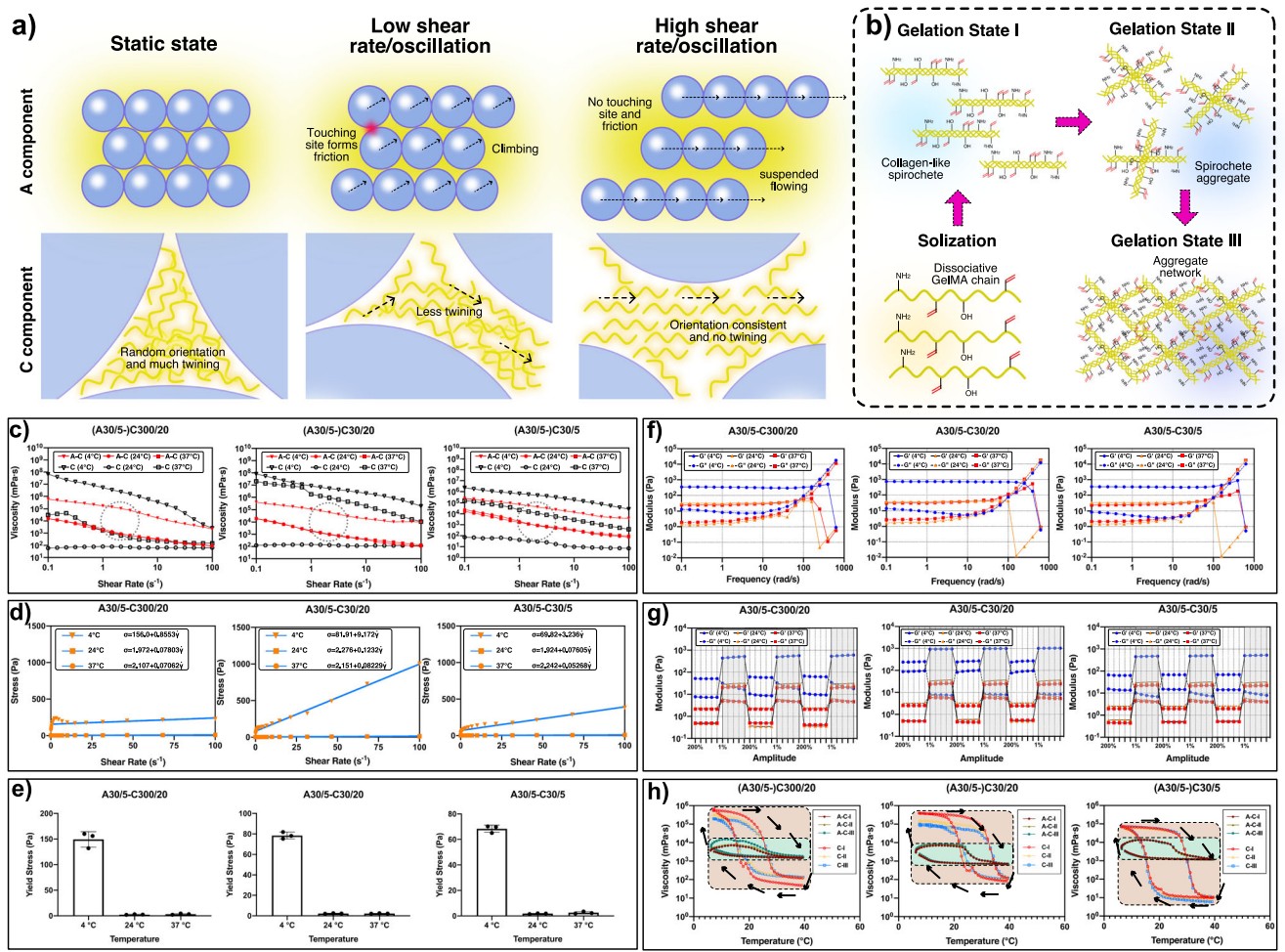

**Fig. 3 Rheological robustness testing of A–C bioink. a** The mechanism of the shear-thinning feature of A–C bioink. **b** The mechanism of the GelMA precursor solution sol–gel transferring. **c** Flow sweep testing. **d** Fitting with Bingham fluid model. **e** Yield stress. ($n = 3$ independent experiments. Data are presented as mean value ± SD.) **f** Oscillation frequency testing. **g** Thixotropy testing. **h** Recycled flow temperature ramp testing.

maintained inside 1 magnitude because of the dominant role of microgels. Remarkably, promising bioprinting temperatures of 20–24 °C were also the dramatical viscosity-changing range. It could be imagined that once a tiny temperature floating in this range happened during the in-situ bioprinting, the viscosity of traditional bioink would sharply vary and bring unpredictable risk while A–C bioink can perfectly maintain the viscosity stability to a great extent and guarantee the treatment validity.

**Printability in simulated in-situ bioprinting.** A simulated in-situ bioprinting scene was established to evaluate the printability of A–C bioink from the extruding and deposition states. For extruding state, three environment temperatures (4, 24, 37 °C) and A30/5–C300/20 and C300/20 were selected and extruded at a constant flow rate. (Fig. 4a). At 37 °C, C bioink was in an excessive solization state and form droplets. At 4 °C, C bioink was in an excessive gelation state and had become hydrogel bulk in the syringe, intermittently forming fragment. C bioink showed good printability only at 24 °C and formed uniform filaments. However, A–C bioink, thanks to its great rheological robustness, could form uniform filaments at the three temperatures. For deposition state, the environment temperature was set as 24 °C to achieve the best extruding state. To simulate the patients' wounds, the receiving platform was set as 37 °C (body temperature) and some food coloring solution (blood) was daubed (Fig. 4b). The deposited A–C bioink could perfectly maintain the 3D structure

while C bioink gradually melted and mixed with the "blood". Thus, A–C bioink would show great adaptation to the complex conditions around wounds. Moreover, A–C bioink was successfully printed with a commercial 3D bioprinter to establish a 3D cube with 12 layers and 7.5 mm height in the roughly controlled environment temperature (30 °C) and on the "wound" receiving basement. (Fig. 4c, Supplementary Notes 5, 9 and Supplementary Video 3).

**Binding effect on the defect–hydrogel interface.** In in-vitro bioprinting, 3D structures are crosslinked on the deposition platform, resulting in the lack of strong binding force with tissues after transplanting. During therapy, the shifting of the transplanted structures can be invalid or dangerous for patients. By contrast, in-situ deposited A–C bioink would form a strong binding force on the defect/hydrogel interface (Fig. 4d). This is because C component can display fluidity after contacting body temperature and easily infiltrate the defect vacancy, increasing the attaching sites on the interface and the internal friction (Fig. 4e). Furthermore, based on the special chemical properties of GelMA, A–C composite structures can build auxiliary interface force on the wound, which was probably due to photo-generated aldehyde groups bonding with the amino groups on the tissue surface[34] (Fig. 4f). To distinctly observe the strong binding force, A30/5–C30/20 bioink was poured on fresh pig tendon (Fig. 4g), while A30/5–C300/20 bioink was poured on fresh pig rib (Fig. 4i).

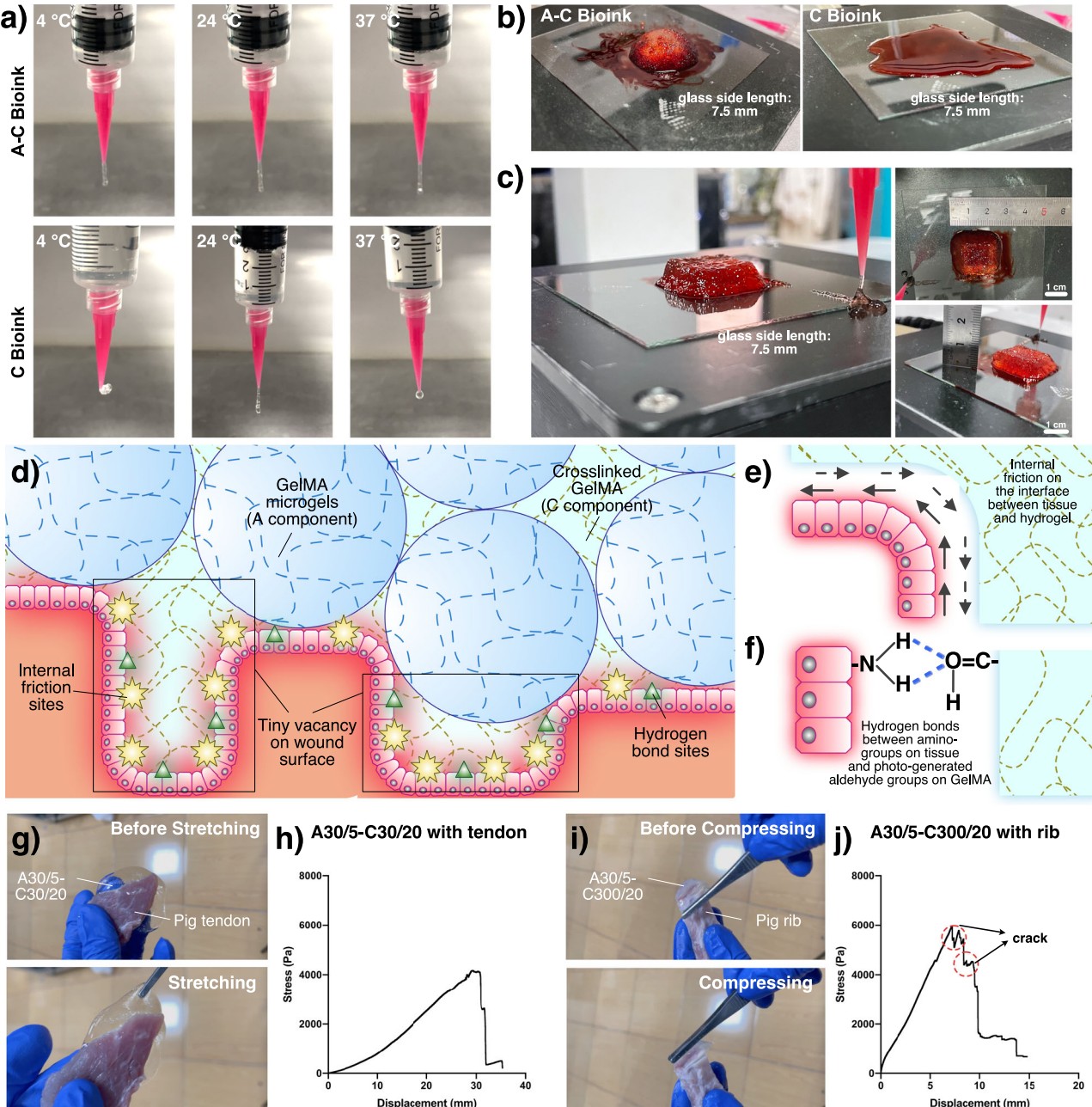

**Fig. 4 In-situ bioprinting simulation on traditional 3D bioprinter and binding force. a** The extruded filament shapes in different temperature. **b** Shape maintaining at "wound" filled with "blood" at 37 °C. **c** 3D printing of cube at "wound" filled with "blood" at 37 °C with A–C bioink. **d** Mechanism of the stable binding force forming. **e** Internal friction on the interface. **f** Hydrogen bonds between amino-groups on tissue and photo-generated aldehyde groups on GelMA. **g** Tensile binding testing with A–C bioink and pig tendon. **h** Binding stress test of A30/5–C30/20 with pig tendon. **i** Compressive binding testing with A–C bioink and pig rib. **j** Binding stress test of A30/5–C300/20 with pig rib.

All-direction forces were added to A–C structures (Supplementary Video 4). A–C structures strongly attached to the tissue surface. To explore the binding force between A–C bioink and tissue surface, A30/5–C30/20 and A30/5–C300/20 were poured (about 2 mm height) and photocrosslinked between two pieces of fresh pig tendon (cross-section was about 3 cm × 1 cm) and two pieces of fresh pig ribs (cross-section was about ⌀1.6 cm), respectively. The binding stress was tested by the method of stretching the tissue–hydrogel–tissue structure by clipping two pieces of tissue in the opposite direction at 1 mm/min. The binding stress between A30/5–C30/20 and pig tendon could reach above 4000 Pa (Fig. 4h) and the one between A30/5–C300/20 and pig rib could reach nearly 6000 Pa (Fig. 4j). Two steps occurred on A30/5–C300/20 and rib curve because A30/5–C300/20 was more suitable for compressing and would form crack during the stretching test. All the results indicated that A–C bioink would meet the requirement of strong binding force.

**Mechanical properties of A–C composite structure.** Compared to the traditional method for establishing composite structures, namely, printing strengthen scaffolds followed by casting soft hydrogel, the method based on A–C bioink has obvious superiority. Firstly, the structure design can be more flexible and strengthen scaffold network would spontaneously form around A component. Furthermore, because A/C component are both

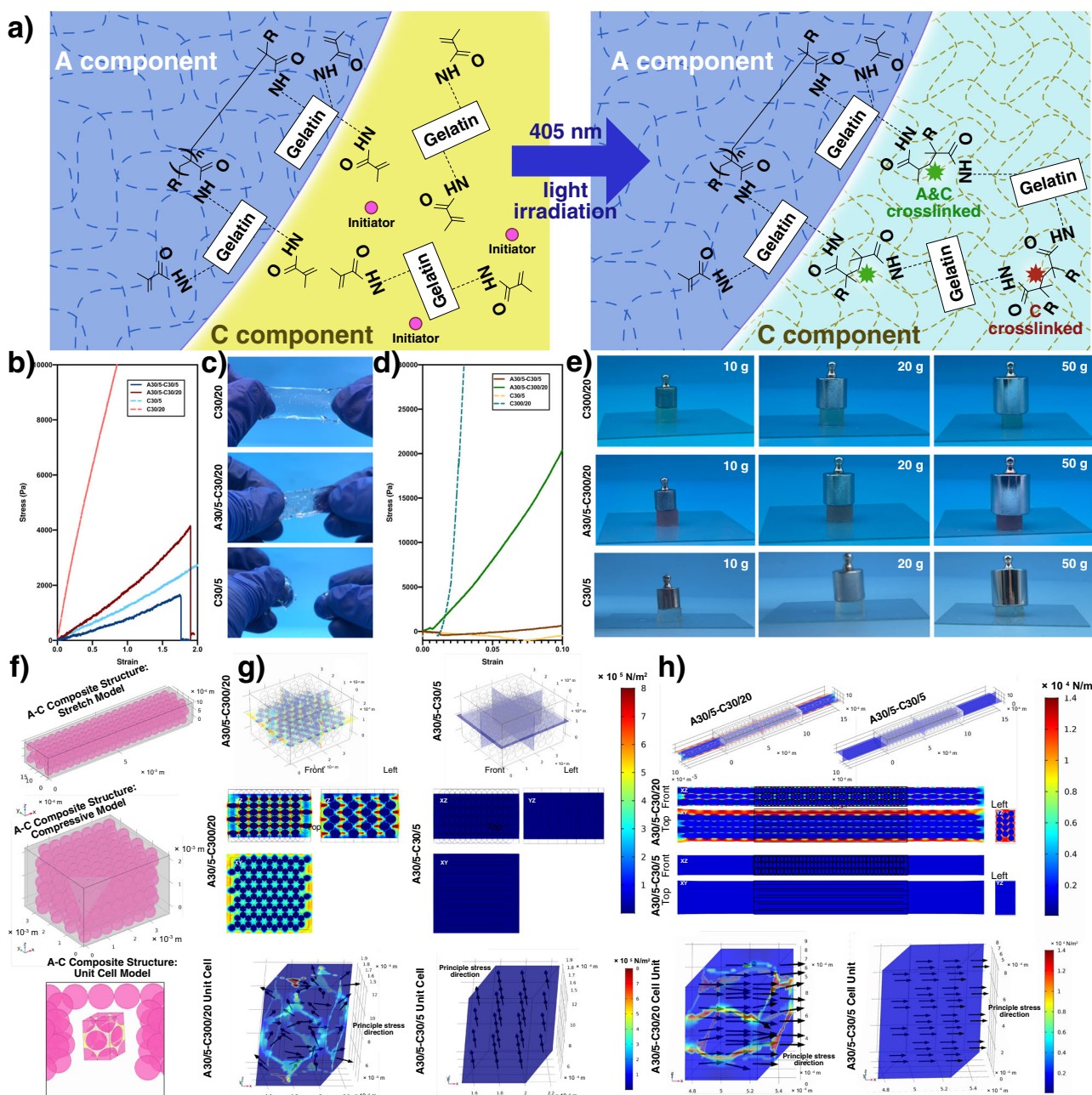

**Fig. 5 Mechanical properties of A–C composite structure. a** Mechanism of the stable covalent bonds forming process of A–C composite structure. **b** Tensile testing curve. **c** Tensile state of different bioink. **d** Compressive testing curve. **e** Compressive state of different bioink. **f** Simulation models of tensile/compressive A–C composite structure and unit cell. **g** Compressive simulation of A–C composite structure. **h** Tensile simulation of A–C composite structure.

GelMA possessing carbon–carbon double bonds, during photo-crosslinking, the unreacted carbon–carbon double bonds on the surface of GelMA microgels in electrospraying would break up and connect with the broken ones in C component, forming strong covalent bonds on A/C interface (Fig. 5a), which is absent in traditional methods.

The compressive properties of A30/5–C300/20, C300/20, A30/5–C30/5, C30/5 and tensile properties of A30/5–C30/20, C30/20, A30/5–C30/5, C30/5 were tested. The compressive modulus was 204.00, 2608.00, 16.26, and 3.73 kPa, respectively (Fig. 5b) and the Young's modulus was 1.81, 11.98, 0.80, and 1.26 kPa, respectively (Fig. 5d), indicating C component obviously strengthened the mechanical properties of the printed structure, which was also proved by the visual observation (Fig. 5c–e).

The stress distribution inside A–C composite structure was analyzed with finite-element simulation. The structure was simplified as an HCP model (Fig. 5f). The displacement boundary condition in the tensile/compressive model was set as 50%/10%, respectively (Fig. 5g, h). Compared to C structure, high Von Mises stress distributed among the microgels in A–C composite structure, just like a strong scaffold "printed" inside, which made it possible to provide extracellular matrix (ECM) with similar mechanical environment in different A–C bioink types. By analogy with the unit cell research method in crystal chemistry, we innovatively set up the concept of "A–C unit cell" with the same incision method as HCP, displaying regular high-stress network packed the low-stress microgels inside both in compressive/tensile models.

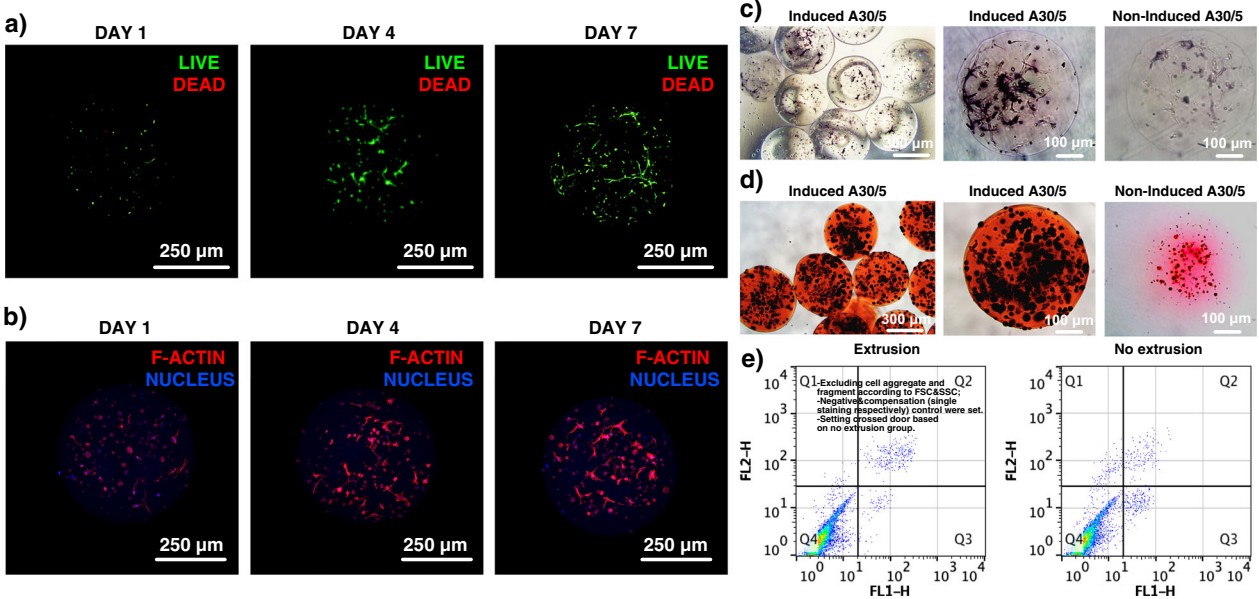

**Fig. 6 Culturing and osteogenic induction of BMSCs-laden A component. a** Viability of the BMSCs encapsulated in GelMA microgels. **b** Actin morphology of the BMSCs encapsulated in GelMA microgels. **c** ALP testing of the osteogenically inducted BMSCs-laden A component. **d** ARS testing of the osteogenically inducted BMSCs-laden A component. **e** Apoptosis testing with Annexin V-FITC/PI by flow cytometry. For **a–d**, each experiment was repeated independently with similar results for 3 times.

**Osteogenesis of BMSCs in A component**. A30/5 encapsulating bone marrow stromal cells (BMSCs) were electrosprayed and cultured to be further used as functional cell therapy units in A–C bioink. The cell viability on 1st, 4th, and 7th day showed to be above 90% with the LIVE/DEAD kit (Fig. 6a), indicating the basic biocompatibility of A30/5 and F-actin morphology displayed that BMSCs could gradually spread inside the 3D microenvironment (Fig. 6b). Furthermore, some A30/5 encapsulating BMSCs were cultured in osteogenic induction medium after three-day basic culturing. On the 7th day of induction, A30/5 was stained with alkaline phosphatase (ALP) and showed that the induced BMSCs had entered the early osteogenic stage (Fig. 6c). On the 21st day of the induction, A30/5 was stained with Alizarin Red S (ARS) and showed that the induced BMSCs had entered the late osteogenic stage and produced calcium nodules inside the A30/5 (Fig. 6d), which verified its osteogenic differentiation ability and potential therapy effect. Moreover, A30/5 with BMSCs would experience shear force from printing nozzle during in-situ printing, which could cause cell apoptosis. From the results of apoptosis testing with flow cytometry (Fig. 6e), apoptosis caused by extrusion were not obvious, demonstrating the soft environment formed by A30/5 could protest the encapsulated BMSCs from shear force during extrusion and guarantee the biological function.

**Bone regeneration in cranial defects**. To examine the therapeutic action of A–C bioink, A30/5–C300/20 bioink with or without BMSCs was deposited directly inside the rat cranial defect (column: diameter 5 mm and height 1.5 mm) and photocrosslinked (Fig. 7a). At 2nd week, micro-computed tomography revealed new bone formation in BMSC-loaded A–C group (Fig. 7c). However, no obvious new bone was formed in blank group, and BMSC-unloaded group showed a very limited amount of bone regeneration. It was probably because hydrogels acted as a scaffold for relevant cells on the original defect location and provided more growing space. Furthermore, BMSC-loaded A–C group exhibited better bone regeneration efficacy with a higher BV/TV (Fig. 7b). At 4th week, the bone almost completely bridged the injured site in BMSC-loaded A–C group, and BMSC-unloaded group also exhibited more new bone tissue growth. The BV/TV values increased with time in all groups and both BMSC-loaded and BMSC-unloaded groups were significantly higher than that of blank group. Consistent with radiographic examination, histological analysis with hematoxylin and eosin (Fig. 7d) and Masson trichrome staining (Fig. 7e) at 2nd weeks revealed new bone formation with the largest surface area in the BMSC-loaded group sample, and no marked bone formation in blank and BMSC-unloaded groups. At 4th week, bone formation increased in all groups. Histological observations at higher magnifications confirmed that the neo-formed bone with typical structure in BMSC-loaded group showed many more regions of new mature bone formation than the other groups, indicating BMSC-loaded A–C bioink could promote endogenous bone formation in a critical-size rat cranial defect model.

**In-situ cranial repair of different defect morphologies**. The actual clinical cases of organ defect are caused by all kinds of accidents, such as fires, traffic accidents, and military injuries. Thus, the 3D morphologies and sizes of the organ defects are very different. To examine the in situ bioprinting capability of A–C bioink in a clinical setting, four rat "patients" with cranial defects with approximately rectangular, square, trapezoid, and triangular shapes (1.5 mm height) were created with a dental trephine (Fig. 8a–c). The robotic arm system was selected as the in-situ bioprinting tool (Fig. 8b). Four rat "patients" were placed on the operating table. The 3D models of the defects were rebuilt with computer-aided design software, and the printing routine program was generated with slicing software and loaded into the controlling system of the robotic arm. A30/5–C300/20 bioink encapsulated BMSCs were in-situ deposited into the cranial defect of four "patients" and photocrosslinked with 405-nm blue light (Supplementary Video 5). After 6 weeks of implantation, micro-computed tomography revealed that new bone was formed from the edge toward the center of the defects in all "patients" (Fig. 8d, e), which verified the high feasibility of A–C bioink in in-situ bioprinting therapy.

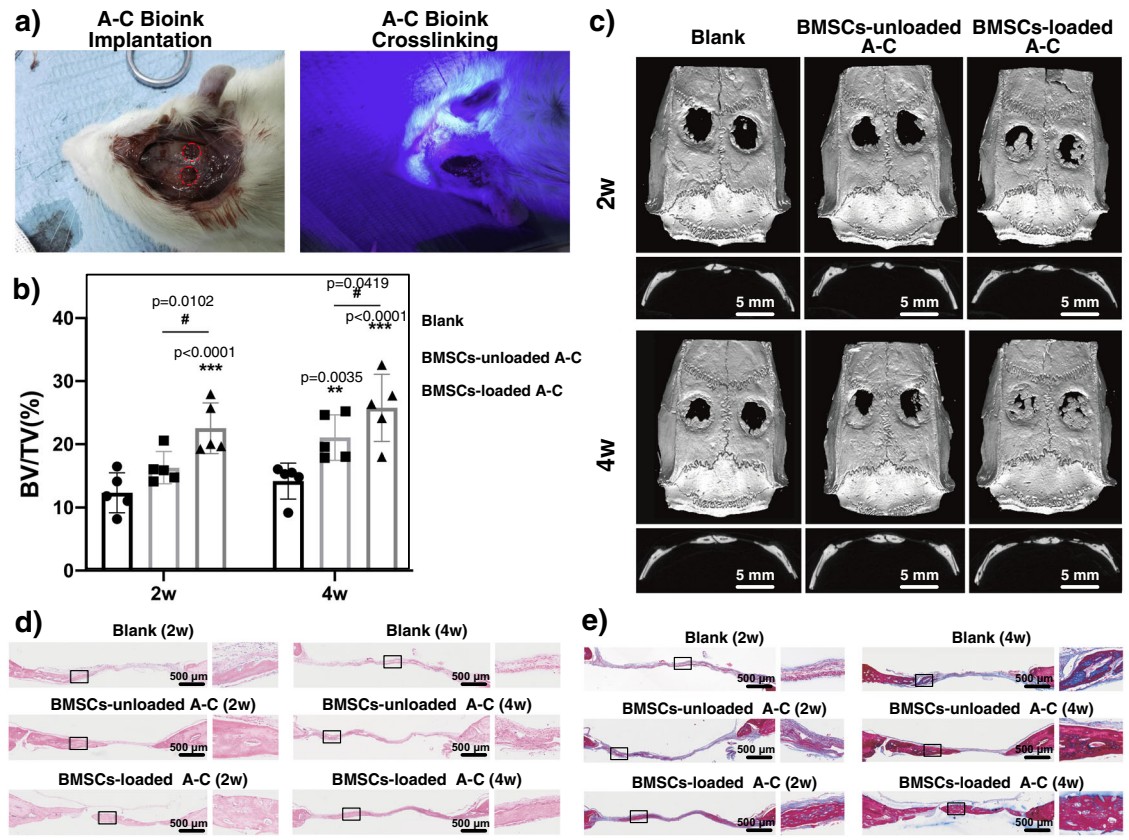

**Fig. 7 In-vivo therapeutic effect of A–C bioink at rat cranial defect model. a** Implantation and photocrosslinking of A–C bioink at rat cranial defect model. **b** BV/TV value. ($n = 5$ rats. Data are presented as mean value ± SD with GraphPad Prism 8.) Two-sided two-way ANOVA followed by Tukey post hoc multiple comparisons test was used. And statistical test was conducted within 2w groups or 4w groups, respectively, but not done across 2w and 4w groups. **c** Micro-computed tomography examination. **d** Histological analysis with H&E staining. **e** Histological analysis with Masson trichrome staining. For **d**, **e**, each experiment was repeated independently with similar results for 3 times.

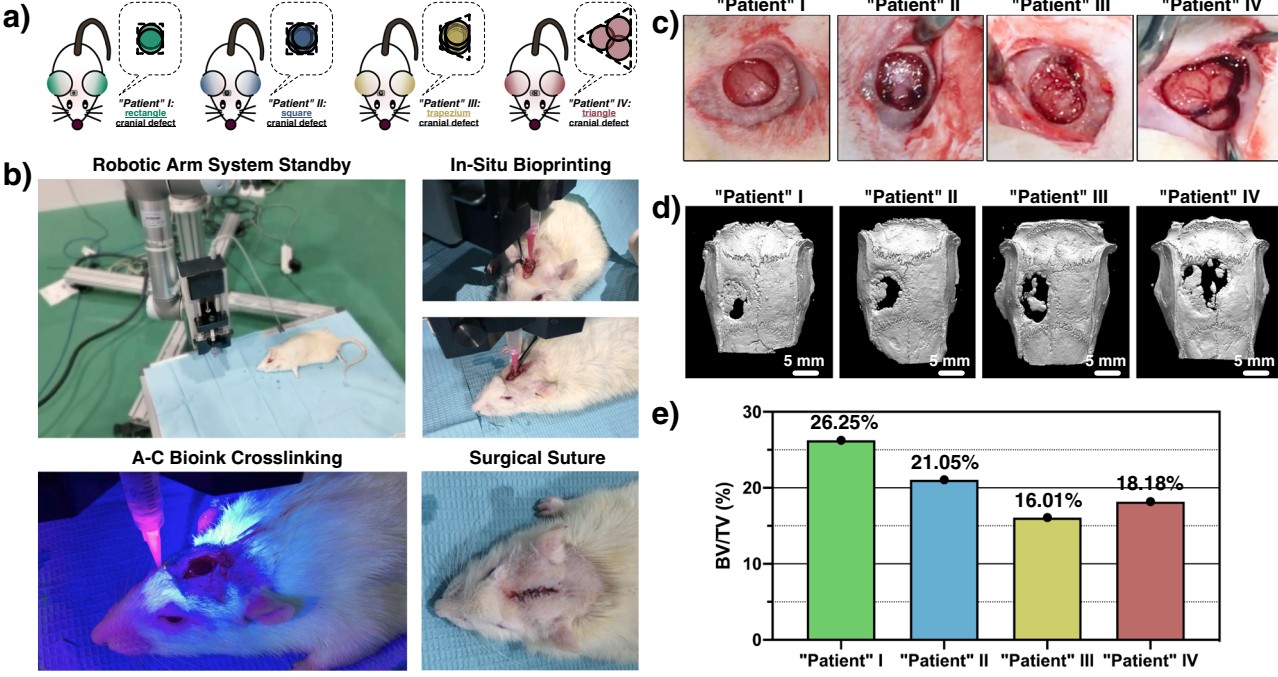

**Fig. 8 In-situ bioprinting at rat cranial defect models with different morphology with A–C bioink. a** 3D structure morphologies of the rat cranial defect models. **b** In-situ bioprinting steps with A–C bioink. **c** Original "patient" cranial defect morphology. **d** Micro-computed tomography examination after 6-week in-situ treatment. **e** BV/TV value after 6-week in-situ treatment.

## Discussion

Most of the current studies of microgel-based bioink only deposited gathered microgels in vitro/vivo to form 3D structures, which can collapse or be pushed out of the wound site hypodermically with the daily activities of patients. All the while, the excellent features and revolution of microgel-based bioink have been ignored in in-situ bioprinting. The bioconcrete bioink proposed here contained an on-demand cement component, namely, a different hydrogel precursor solution in the bioink system, and the interaction between the aggregate and cement component successfully solved the problems which have been restricted in the development of clinical in-situ bioprinting. By combining the promising rheological properties of microgels and the mechanical adjustability of GelMA, A–C bioink displayed good in-situ printing robustness, simplicity of composite structure establishment, in-vivo therapeutic action, binding force on tissue/hydrogel interface and portability.

Thus far, more and more fabrication methods of functional microtissues have been proposed. Besides, the cell-laden A component can be easily pre-inducted with specific induction medium and stored in liquid nitrogen for better portability and therapeutic effect, making it possible to realize the mass production and transportation of A–C bioink to meet the clinical requirements. We believe that A–C bioink will act as a significant role in organ defect treatment and make a big push to develop clinical in-situ bioprinting technology.

In the future, A–C bioink can be designed in more ways. For A component, cell species in A component can be changed for more functions (Supplementary Note 7). Besides, hierarchical microgels were fabricated in our previous study[35], with which synergistic treatment actions could be realized. For C component, GelMA is the only biomaterial applied here and other biomaterials could be as substitutes such as hyaluronic acid methacryloyl (HAMA) with high mechanical adjustability[36–40]. Furthermore, in exploration, we found A component encapsulating endothelial cells could be vascularized by C component encapsulating tumor cells secreting vascular endothelial growth factor (VEGF)[41,42] (Supplementary Note 8), which would probably realize in-situ vascularization on organ defects. Moreover, nutrition/gas supplying problem in large-scale hydrogel structure is a common problem in either in-situ bioprinting or in-vitro bioprinting. Fortunately, researchers in our lab[43] or other labs[44] have proposed a series of effective way to solve this problem. Therefore, in future, next generation A–C bioink could be designed as the one with sacrificed component or phase separation component to form more nutrition network in the A–C structure.

Considering the portability and multi-scene feasibility, we call on the development of intelligent and carry-home equipment for A–C bioink in-situ bioprinting in the future. As an assumption, a compositive bottle integrated heating device, portable battery and flashlight could be designed. Furthermore, from an urban development standpoint, "nitrogen stations" like petrol stations and "shared handheld in-situ bioprinters" like shared bicycles can be established at public positions as social service, so that the long-term storage of A–C bioink in long journeys and the immediacy of in-situ bioprinting can be guaranteed.

## Methods

**Electrospraying of GelMA microgels (A component)**. Pure 5% (w/v) GelMA prepolymer solution was prepared by dissolving the freeze-dried GelMA (EFL-GM-30, purity > 99.9%) and lithium phenyl-2, 4, 6-trimethylbenzoylphosphinate (LAP, 0.5% (w/v), purity > 99.8%) in phosphate buffered saline (PBS). The solution was filtered through a 0.22-μm filter. An electric field was formed with the metal nozzle (30 G) and metal ring. The flow rate of the electrospraying ink was set as 50 μL/min and driven by an injection pump. The voltage was set as 2.86 kV. The environment temperature was set as 30 °C to ensure the suitable fluidity of the electrospraying ink. The electrosprayed microdroplets were received by a Petri dish filled with

silicon oil and crosslinked by 405-nm blue light. The crosslinked GelMA microgels were transferred to a centrifugal tube and centrifugated at $128.57 \times g$ for 5 min (3 times) to remove the silicon oil. The microgels were stored in PBS. For the BMSC-laden GelMA microgels, BMSCs were mixed in the electrospraying ink at a cell density of $5 \times 10^5$ cells/ml. The prepared BMSC-laden GelMA microgels were cultured in DMEM/F-12 complete medium supplemented with 10% (v/v) fetal bovine serum (FBS) at 37 °C and 5% $CO_2$.

**Preparation of C component**. Pure GelMA prepolymer solution was prepared by dissolving the freeze-dried GelMA (EFL-GM-30/EFL-GM-300, purity > 99.9%) and lithium phenyl-2, 4, 6-trimethylbenzoylphosphinate (LAP, 0.5% (w/v)) in phosphate buffered saline (PBS). The solution was filtered through a 0.22-μm filter.

**Rheological testing of A–C (C) bioink**. For the rheological testing of A–C bioink, a 25-mm parallel plate rotor was selected, and the testing space was set as 2.5 mm (approximately 5 times of the diameter of A component). For the rheological testing of C bioink, a 25-mm parallel plate rotor and 1 mm testing space were set in the 4 °C test and a 50-mm parallel plate rotor and 0.5-mm testing space were set in the 24 and 37 °C tests. For flow sweep testing, the shear rate range was set as $0.1–100 \, s^{-1}$. The data fitting with the Bingham fluid model was carried out with MATLAB. For oscillation frequency testing, the amplitude was set as 1%, and the frequency range was set as 1000–0.1 rad/s. For thixotropy testing, the periodically varied oscillation amplitude (200% and 1%) was added to the samples. The alternation period was set as 30 s, and five points were examined in every stage and repeated three times. For flow temperature ramp testing, the bioink experienced cooling/heating three times back and forth. The temperature changing rate was set as 5 °C/min.

**Mechanical analysis of A–C composite structure**. For the compressive samples, A30/5–C300/20, C300/20, A30/5-C30/5, and C30/5 bioink was poured in the cylindrical mold (φ9 mm × 6.3 mm) and crosslinked with 405-nm blue light. For the tensile samples, A30/5–C30/20, C30/20, A30/5–C30/5, and C30/5 bioink was poured in the cuboid mold (20 mm × 3 mm × 5 mm). The mechanical testing was carried out on a hydrogel sample testing machine. The motion rate was set as 1 mm/min. The mechanical simulation was executed with COMSOL Multiphysics. The Poisson's ratios of C30/20 and C30/5 structures were set as 0.034 and 0.031, respectively. The displacement boundary condition in the tensile simulation was set as 50% of the original length of the model. The Poisson's ratios of C300/20 and C30/5 structures were set as 0.218 and 0.031, respectively. The displacement boundary condition in the compressive simulation was set as 10% of the original length of the model.

**Basic culturing and staining of BMSC-laden A component**. BMSCs (primary cells from rats) were provided by Stomatology Hospital, School of Stomatology, Zhejiang University School of Medicine, Zhejiang Provincial Clinical Research Center for Oral Diseases, Key Laboratory of Oral Biomedical Research of Zhejiang Province, Cancer Center of Zhejiang University, Hangzhou 310006. The electrosprayed BMSC-laden GelMA microgels were cultured in DMEM/F-12 complete medium. BMSC viability was measured after 1, 4, and 7 days. The cell viability was tested with the Live/Dead assay for 40 min. Thereafter, CLSM was applied to image the encapsulated BMSCs by obtaining two images of each frame: green for live cells and red for dead cells. The live and dead cell numbers were analyzed with ImageJ, and the BMSC viability was calculated as the ratio of the number of live cells to the total number of cells. For the morphology of the encapsulated BMSCs, they were stained with a cytoskeletal dye, including actin which with stained with TRIC phalloidin (40734ES75, Yeasen Biotechnology), and the nucleus was stained with DAPI. The BMSC-laden A component was imaged with CLSM.

**Induction and staining of BMSC-laden A component**. The electrosprayed BMSC-laden GelMA microgels were cultured in DMEM/F-12 complete medium for 3 days. Thereafter, DMEM/F-12 osteogenic induction medium was prepared with dexamethasone (0.1 mM), β-glycerol phosphate disodium salt hydrate (10 mM), and L-ascorbic acid (50 μg/mL). The BMSC-laden GelMA microgels were inducted with the prepared induction medium. On the 7th day, the BMSC-laden A component was fixed with 4% (w/v) polyformaldehyde for 30 min and stained with alkaline phosphatase (ALP). On the 21st day, the BMSC-laden A component was fixed with 4% (w/v) polyformaldehyde for 30 min and stained with Alizarin Red S. The stained samples were observed with an optical microscope.

**Apoptosis testing during extrusion**. The A30/5 with BMSCs were electrosprayed as above, half of which were extruded from 20 G cone-shape nozzle at 150 μL/min. After 4-h culturing, the extruded and non-extruded A30/5 with BMSCs was degraded with 20 U/mL collagenase II PBS solution for 30 min (37 °C) to remove GelMA hydrogel. The harvested cells were stained with Annexin V-FITC/PI kits and tested by flow cytometry, respectively. The data were analyzed with FlowJo software.

**Rat calvarial defect model building and implantation**. Thirty 12-week-old male SD rats (250–300 g) were provided by the Animal Experiment Center of Zhejiang University (Hangzhou, China). The experiments were approved by the Ethics Committee for Animal Research at Zhejiang University (ethics approval number: ZJU20210172) and performed in accordance with the Institutional Animal Care and Use Committee of Zhejiang University. The SD rats were randomly assigned to groups, including blank, blank-concrete, and induced-concrete hydrogel microspheres, to evaluate the osteogenic potential in a cranial defect on the implantation of hydrogel microspheres. SD rats were anesthetized with an intraperitoneal injection of 2% pentobarbital sodium. After complete shaving and disinfection, a longitudinal incision was made in the middle of the surgical area, and then the soft tissue was carefully separated to expose the calvarium. The periosteum was stripped out, and two bilateral defects that were 5 mm in diameter were created with a dental trephine. The hydrogel microspheres were then implanted into the defects. Penicillin was injected once a day postoperatively for three days.

**Micro-computed tomography analysis**. After 2 and 4 weeks of implantation, the rats ($n = 5$ in each group) were euthanized by $CO_2$ suffocation, and the calvarial specimen was harvested and fixed in 4% (w/v) paraformaldehyde for further characterization. The three-dimensional (3D) structures of the regenerated bone tissue within the cranial defect area were evaluated with micro-CT (SkyScan, SkyScan 1176, Belgium) with the following scanning parameters: a resolution of 18 μm, voltage of 65 kV, current of 385 μA, and A1 filter of 0.5 mm. The 3D reconstruction was performed with system software (SkyScan, Belgium). The ratio of bone volume to tissue volume (BV/TV) was quantitatively determined with a cylinder ROI of 5 mm in diameter.

**Histological analysis**. After 2 and 4 weeks of implantation, the harvested specimens were fixed in 4% (w/v) paraformaldehyde for 48 h and decalcified in 15% ethylene diamine tetraacetic acid (EDTA) solution for 2 weeks at room temperature. The EDTA solution was refreshed every 3 days. The samples were then dehydrated through a graded alcohol series and embedded in paraffin. Histological sections with an approximate thickness of 5 μm were obtained from the center of the embedded specimens, followed by hematoxylin and eosin (H&E) or Masson trichrome staining to assess bone formation. Images were obtained under a bright field microscope (Olympus, Tokyo, Japan).

**In-situ bioprinting with A–C bioink on rat calvarial defect**. The different cranial defect morphologies were created with a dental trephine (5 mm diameter, 1.5 height). The defect morphologies of "patients" I, II, and III were rectangular, square, and trapezoid in shape, respectively, with 2, 4, and 5 circles, and the center distance was 1 mm. The defect morphology of "patient" IV was triangular in shape with 3 circles, and the center distance was 4 mm. The defect models were rebuilt in Solidworks and sliced with Repetier software to acquire the path information, followed by transfer to the corresponding controlling program that used the controlling software of the robotic arm system. The bioink flow rate was set as 150 μL/min, and the nozzle movement speed was set as 120 mm/min. The 20G cone-shape nozzle was selected as the printing nozzle. The $X$–$Y$ original point of the nozzle was set according to the back right edge of the cranial defect, and the $Z$ original point was set according to the the highest point of the skull around the defect. The injection pump was fixed on the end of the robotic arm system. After A–C bioink printing, the deposited bioink was crosslinked with 405-nm blue light for 30 s, and the scalp was sutured and sterilized.

**Reporting summary**. Further information on research design is available in the Nature Research Reporting Summary linked to this article.

## Data availability
The authors declare that all data supporting the findings of this study are available within the paper and its Supplementary Information or from the corresponding authors upon request.

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

## Acknowledgements

This study was sponsored by the National Key Research and Development Program of China (2018YFA0703000, Y.H.), the National Natural Science Foundation of China (No. U1909218, Y.H.), the Science Fund for Creative Research Groups of the National Natural Science Foundation of China (No. T2121004, Y.H.).

## Author contributions

M.X. performed all of the fabrication and in-vitro experiments and wrote the paper with input from all the authors. Y.S. performed the animal experiments and characterization. C.Z. and M.G. assisted to perform the animal experiments and characterization. J.Z. assisted in the design of in-situ bioprinting path program and operation of robotic arm. J.F., Z.C. and Z.X. contributed to the study design. Y.H. organized the project and gave project suggestion.

## Competing interests

The authors declare no competing interests.
