## [Peer Review File · Nature Communications]

In Situ 3D Bioprinting with Bioconcrete BioinkREVIEWER COMMENTS

Reviewer #1 (Remarks to the Author):

In this article, the author constructs a heterogeneous bio-ink composed of cell-loaded microgels. The non-uniform hydrogel has perfect Bingham fluid characteristics. At body temperature, the prepared bio-ink exhibits good printability and stability. The article is very innovative. But the author still needs to pay attention to the following issues.

Major concerns

1. The diameter of the microspheres fabricated by spraying is not uniform. The stacked structure of those microspheres is irregular. Therefore, it is inappropriate to use the unit cell structure in solid chemistry as an analogy (the same element in the unit cell has the same atomic radius).
2. The author mentioned in the article (lines 107-108) that GelMA hydrogel can withstand bone pressure and tendon tension. However, the tensile strength (or young's modulus) and compressive strength (or compressive modulus) of GelMA hydrogel are too different from these structures. In fact, GelMA hydrogel does not have such high mechanical properties.
3. Does this heterogeneous bio-ink derived from microspheres affect printing accuracy? The author should provide more data for this.
4. When the bio-ink passes through the needle, the shearing force suddenly increases. It increases the instability of the cell membrane. Some sheared cells lose the ability to proliferate, even if they are still alive. The author should examine the proportion of apoptosis after being squeezed by the nozzle.
5. In Figure 4 g, the author believes that the photo-generated aldehyde group and the microsphere structure can fit well with the irregular wound surface of the damaged tissue. The author should quantify the tear strength between the hydrogel-tissue interface.
6. The microgel-derived bio-ink shows excellent mechanical properties compared to pure GelMA hydrogel. Due to the different mechanical properties of the microgel and the surrounding GelMA gel, a mechanical gradient will be formed inside and outside the microgel after crosslinking. It is well known that stem cells show different differentiation in hydrogels with variable modulus and structures. This may cause the cells in the hydrogel to differentiate into different cells under the same physiological conditions, thereby forming a heterogeneous tissue. The author should discuss the potential risk of this in the article.

Minor concerns

1. The enlarged images of Figure 2b and Figure 2c are not clearly marked, which is easy to cause misunderstanding by readers.
2. Figure 3e should be measured 3 or 5 times to get data with error bars.
3. Figures 2e and f should have a scale bar.

Reviewer #2 (Remarks to the Author):

This manuscript mainly described a novel bioconcrete bioink with electrosprayed cell-laden microgels as the aggregate and gelatin methacryloyl (GelMA) precursor solution as the cement, which could be used for in-situ bioprinting, composite structure establishment, and bone regeneration.

In my opinion, this article is complete in structure and clear in organization. However, some discussion in the article may be vague, and the corresponding explanations need to be supplemented. In addition, there may be problems with individual figures in the article. What's more, this manuscript is rich in content and has a certain reference value in the 3D bioprinting field. So, I suggest that this article can be published, but the paper needs minor revision before acceptance for publication. My detailed comments are as follows:

1. In this paper, the 405 nm blue flashlight was used for photo crosslinking. So, what way could you evaluate the damage of this light used to biological tissues?
2. In Figure 4f, the binding bonds should be characterized, such as photo-generated aldehyde groups on GelMA.
3. The figures including Figure 3 and Figure 5 are too small and the error bar in Figure 7 is blurred.
4. In Emergency bag for portability, the freezing tubes could be directly heated using the body temperature. But what way could avoid the cold injury on the skin?
5. In addition to 405 nm blue flashlight, is it possible to consider using other light sources that are safer for this printing, such as green light and red light.
6. In Figure 4b-c, the scale bar should be added.

Reviewer #3 (Remarks to the Author):

In this study, the authors propose in situ bioprinting with a bioink composed of Gelma microgels and a Gelma precursor solution. The authors show that by reducing the volume ratio of the precursor solution to the microgels as much as possible, the bioink could maintain its shape after being ejected from a nozzle, as observed in the case of the Brigham fluid. In addition, the authors show that the bioink strongly bonds onto a recipient tissue through reactions with the photo-generated aldehyde groups. The authors further demonstrate that BMSC-encapsulated bioink is effective for bone regeneration in a rat cranial defect model. Although the manuscript includes vital data, certain results and descriptions are unclear, and a few of the interpretations are debatable. Furthermore, the conclusions might be partially unsupported or insufficiently proven by the results.

The critical issue in this study is that no data is provided on the thickness of the tissue that can be treated with their approach. As there are no vasculatures or microchannels in the printed constructs, the supply of oxygen and nutrients to the embedded cells in thicker tissue grafts can cause significant complications. These issues become further apparent while printing tissues with a higher cell density similar to the in vivo tissues, although the present study used microgels with a low cell density (Fig. 6c) for the treatment of the thin tissues, such as defects of rat cranial bones. The emergency bag is shown in Fig. 2; however, the volume of the bioink seems to be small, and it is unclear where and how large of a tissue defect can be treated using that. In addition, the authors state in the entire manuscript including the title that the key point of this study is in situ 3-D bioprinting. However, there is no data showing that bioprinting with their bioink is beneficial. The authors conducted in situ bioprinting of the bioink onto four different cranial bone defects. In the experiments, the bone generation should have been compared to a control experiment in which the bioink is manually loaded with a pipette to demonstrate that the bioprinting of their bioink is beneficial. Furthermore, their approach is to use previously photocrosslinked microgels, and therefore the performance should be compared with the typical method of bioprinting during UV irradiation.

In sentence number 116, the authors state that “The volume proportion of A component was analyzed red/green areas as $73.215\% \pm 2.312\%$ (Supplementary Note 2), which was similar to the atomic space utilization in hexagonal closest packed (HCP) in crystal chemistry (74.05%) (Fig. 2d)” However, Fig. 2c shows relatively large areas of red fluorescence and black areas with no fluorescence, which appears to contradict this statement.

The authors emphasize the advantages of in situ bioprinting however concerns or limitations are not mentioned. For example, in situ bioprinting has the disadvantage of irradiating the wound with UV light and the photoinitiator cannot be removed and enters the body.

Answers to reviewers:

Mark Implication:

"Content": Review's remark

"Content": Excerpted content from previous manuscript and supplementary information

"Content": Answer to reviewer's remark

"Content": Revised content in the manuscript and supplementary information

Reviewer #1 (Remarks to the Author):

In this article, the author constructs a heterogeneous bio-ink composed of cell-loaded microgels. The non-uniform hydrogel has perfect Bingham fluid characteristics. At body temperature, the prepared bio-ink exhibits good printability and stability. The article is very innovative. But the author still needs to pay attention to the following issues.

Thanks for your valuable comments on our work. The answers in detail to the proposed questions are aligned below.

Major concerns:

1. The diameter of the microspheres fabricated by spraying is not uniform. The stacked structure of those microspheres is irregular. Therefore, it is inappropriate to use the unit cell structure in solid chemistry as an analogy (the same element in the unit cell has the same atomic radius).

In fact, GelMA microspheres generated by EHD method were very uniform, which has been verified in previous **Supplementary Note 4** and **Supplementary Fig. 6**. The experiment data have been summarized with 50 microgels in each group in the previous version (the diameter differences in each group were too small to show the error bar). Furthermore, we assumed that **Fig.2b-2c** caused reviewer's misunderstanding, in which the microspheres looked not uniform. Actually, it was because the microspheres were in different focal planes relative to the confocal fluorescence microscope lens.

To explained it clearly, we have added the optical image of A30/5 (GelMA microgels electrosprayed by EHD) in **Supplementary Fig. 6b**:

b)

Related description has been added as below:

Volume proportions of A/C component (Manuscript):

"which has been verified to own very uniform diameters (Supplementary Note 4 and Supplementary Fig. 6)"

Supplementary Note 4 (Supplementary Information):

"(50 microgels were measured in each group) in Supplementary Note 4"

"Microgels with diameter of 500 μm were chosen as A30/5 (Supplementary Fig. 6b)."

2. The author mentioned in the article (lines 107-108) that GelMA hydrogel can withstand bone pressure and tendon tension. However, the tensile strength (or young's modulus) and compressive strength (or compressive modulus) of GelMA hydrogel are too different from these structures. In fact, GelMA hydrogel does not have such high mechanical properties.

In fact, A30/5-C30/200 and A30/5-C30/20 are just the representation of tissue with high compress modulus and tensile modulus, respectively, rather than meaning they have reached the high modulus of actual tissue. Fortunately, based on current researching results in bioprinting field, next generation of A-C bioink could be further modified to reach higher mechanical module. For example, as discussed in the previous manuscript, the mechanical properties of GelMA can be tuned by modifying the concentration and the degree of substitution. Thus, the mechanical modulus of C structure can be further increased by changing these two bioink parameters.

The previous related content is shown below:

Volume proportions of A/C component (Manuscript):

"The degree of substitution (DS) of GelMA can be easily modified by changing the ratios of gelatin and methacrylic acid. Higher DS and precursor solution concentration result in higher mechanical properties of the photocrosslinked structures....."

Moreover, GelMA is just a representative photocrosslinkable hydrogel to propose this strategy for solving key problem in clinical in-situ bioprinting. As mentioned in the previous manuscript, more novel photocrosslinkable biomaterials could be chosen as the component of A-C bioink to get wider range of mechanical properties matching the injury organ, such as PCL-MA, HA-MA in our lab (EFL company).

The previous related content is shown below:

Outlook (Manuscript):

".....For C component, GelMA is the only biomaterial applied here and other biomaterials could be as substitutes such as hyaluronic acid methacryloyl (HAMA) with high mechanical adjustability....."

Furthermore, in the research of projection-based 3D bioprinting method in our lab, we also found that different photocrosslinking process could obviously change the mechanical properties of GelMA (*Y Sun, K Yu, J Nie, M Sun, J Fu, H Wang, Y He**, *Biofabrication* 13 (3), 035032, 2021). Thus, in the future, higher mechanical modulus could be realized by using continuously updating biomaterials and special printing process.

3. Does this heterogeneous bio-ink derived from microspheres affect printing accuracy? The author should provide more data for this.

The printing accuracy is very important in in-situ bioprinting. We have added the orthogonal experiment data of filament diameters with different A-C bioink types, printing speeds and printing temperatures in **Supplementary Fig. 13** as parameter choosing reference in in-situ bioprinting with A-C bioink.

Related description has been added as below:

Printability in simulated in-situ bioprinting (Manuscript):

" , Supplementary Note 9"

Supplementary Note 9 (Supplementary Information)

"A30/5-C30/20 and A30/5-C300/20 were extruded from 20G cone-shape nozzle at 150 μ L/min, respectively. The moving speeds of nozzle were set as 80 mm/min, 120 mm/min and 180 mm/min. The bioink temperatures were set as 4 $^{\circ}$ C, 24 $^{\circ}$ C, 37 $^{\circ}$ C. The diameters of extruded layer filaments were measured with optical microscopy (Supplementary Fig. 13). At 80 mm/min, A-C bioink tended to accumulate around the nozzle because of the slow movement of printing nozzle. At 180 mm/min, A-C bioink

tended to be stretched or even snapped accumulate around the nozzle because of the fast movement of printing nozzle. At 120 mm/min, the filament diameters tended to be uniform under all temperatures. Therefore, 120 mm/min was regarded as promising nozzle moving speed for 20G nozzle and 150 $\mu\text{L}/\text{min}$ bioink flowrate."

4. When the bio-ink passes through the needle, the shearing force suddenly increases. It increases the instability of the cell membrane. Some sheared cells lose the ability to proliferate, even if they are still alive. The author should examine the proportion of apoptosis after being squeezed by the nozzle.

The cell apoptosis after the extruding process is a very important indicator. The apoptosis testing results of the laden BMSCs in A30/5 with/without extrusion has been added in **Fig. 6**. The results showed that there was no obvious apoptosis caused by extrusion. This could be explained by the mechanical property of A-C structure, in which A30/5 provide extremely soft environment for the laden cells and the shear stress during extrusion could be low and safe for their survival.

Related description has been added as below:

Osteogenesis of BMSCs in A component (Manuscript):

"Moreover, A30/5 with BMSCs would experience shear force from printing nozzle during in-situ printing, which could cause cell apoptosis. From the results of apoptosis testing with flow cytometry (Fig.6e), apoptosis caused by extrusion were not obvious, demonstrating the soft environment formed by A30/5 could protect the encapsulated BMSCs from shear force during extrusion and guarantee the biological function."

Apoptosis testing during extrusion (Manuscript):

"The A30/5 with BMSCs were electrosprayed as above, half of which were extruded from 20G cone-shape nozzle at 150 μ L/min. After 4-hour culturing, the extruded and non-extruded A30/5 with BMSCs was degraded with 20 U/mL collagenase II PBS solution for 30 min (37 $^{\circ}$ C) to remove GelMA hydrogel. The harvested cells were stained with Annexin V-FITC/PI kits and tested by flow cytometry, respectively. The data were analyzed with FlowJo software."

5. In Figure 4 g, the author believes that the photo-generated aldehyde group and the microsphere structure can fit well with the irregular wound surface of the damaged tissue. The author should quantify the tear strength between the hydrogel-tissue interface.

The qualification testing of the binding force between A-C structure and actual tissue is necessary. The binding force between the surface of A30/5-C300/20&pig rib and A30/5-C30/20&pig tendon have been tested and the results have been added in Fig. 4.

Related description has been added as below:

Binding effect on the defect-hydrogel interface (Manuscript):

"To explore the binding force between A-C bioink and tissue surface, A30/5-C30/20 and A30/5-C300/20 were poured (about 2 mm height) and photocrosslinked between two pieces of fresh pig tendon (cross-section was about 3 cm \times 1cm) and two pieces of fresh pig ribs (cross-section was about ϕ 1.6 cm), respectively. The binding stress was tested by the method of stretching the tissue-hydrogel-tissue structure by clipping two pieces of tissue in the opposite direction at 1 mm/min. The binding stress between

A30/5-C30/20 and pig tendon could reach above 4000 Pa (Fig.4h) and the one between A30/5-C300/20 and pig rib could reach nearly 6000 Pa (Fig. 4j). Two steps occurred on A30/5-C300/20 and rib curve because A30/5-C300/20 was more suitable for compressing and would form crack during the stretching test. All the results indicated that A-C bioink would meet the requirement of strong binding force."

6. The microgel-derived bio-ink shows excellent mechanical properties compared to pure GelMA hydrogel. Due to the different mechanical properties of the microgel and the surrounding GelMA gel, a mechanical gradient will be formed inside and outside the microgel after crosslinking. It is well known that stem cells show different differentiation in hydrogels with variable modulus and structures. This may cause the cells in the hydrogel to differentiate into different cells under the same physiological conditions, thereby forming a heterogeneous tissue. The author should discuss the potential risk of this in the article.

In this research, though different A-C bioinks were prepared and analyzed, BMSCs were encapsulated in the same ECM, namely A30/5 (GelMA microgels generated by 5% w/v EFL-GM-30 solution), which made it possible to provide the same and suitable mechanical environment for cell growth. Rather than gradient pressure distribution, the pressure distribution in A-C structure owned obvious border according to the simulation results displayed in Fig.5. Thus, the mechanical difference could be ignored and the risk could be negligible.

The previous related content is shown below:

Fig.5 (Manuscript):

Related description has been added as below:

Mechanical properties of A-C composite structure (Manuscript):

" ,which made it possible to provide extracellular matrix (ECM) with similar mechanical environment in different A-C bioink types."

Minor concerns:

1. The enlarged images of Figure 2b and Figure 2c are not clearly marked, which is easy to cause misunderstanding by readers.

The figures have been marked.

2. Figure 3e should be measured 3 or 5 times to get data with error bars.

The results of yield stress have been remeasured for 5 times and the results with error bars have been added to **Figure 3e**.

3. Figures 2e and f should have a scale bar.

To show the general size of the emergency bag, we have added main size of the bag and bottle in **Figure 2e** rather than scale bar. In our opinion, because the images here were captured with normal camera and the items in the same images were placed at different distances to the camera lens, scale bar may mislead the real sizes of the items in the same image.

Reviewer #2 (Remarks to the Author):

This manuscript mainly described a novel bioconcrete bioink with electrosprayed cell-laden microgels as the aggregate and gelatin methacryloyl (GelMA) precursor solution as the cement, which could be used for in-situ bioprinting, composite structure establishment, and bone regeneration.

In my opinion, this article is complete in structure and clear in organization. However, some discussion in the article may be vague, and the corresponding explanations need to be supplemented. In addition, there may be problems with individual figures in the article. What's more, this manuscript is rich in content and has a certain reference value in the 3D bioprinting field. So, I suggest that this article can be published, but the paper needs minor revision before acceptance for publication. My detailed comments are as follows:

Thanks for your valuable comments on our work. The answers in detail to the proposed questions are aligned below.

1. In this paper, the 405 nm blue flashlight was used for photo crosslinking. So, what way could you evaluate the damage of this light used to biological tissues?

The biocompatibility of the combination of GelMA hydrogel, photoinitiator LAP and 405nm blue light has been verified to be safe in lots of published contributions (*ACS Appl. Mater. Interfaces* 2018, 10, 8, 6849–6857; Zhou, Feifei, et al. *Biomaterials* 258 (2020): 120287.; Pan, Jie, et al. *Journal of Materials Science: Materials in Medicine* 31.1 (2020): 1-12.; Wang, Guanhuier, et al. *Gels* 7.4 (2021): 247.). Furthermore, 405 nm blue light has been being widely applied in current clinical scenes, such as tooth photocrosslinking, blue light treatment of neonatal jaundice, etc. Therefore, the damage to the laden cells caused by 405 nm blue light can be ignored.

Related description has been added as below:

Emergency bag for portability (Manuscript):

", which has been verified to be safe and widely used in current clinical scenes, such as tooth photocuring, blue light treatment of neonatal jaundice, etc."

2. In Figure 4f, the binding bonds should be characterized, such as photo-generated aldehyde groups on GelMA.

The qualification testing of the binding force between A-C structure and actual tissue is necessary. The binding force between the surface of A30/5-C300/20&pig rib and A30/5-C30/20&pig tendon have been tested and the results have been added in **Fig. 4**.

Related description has been added as below:

Binding effect on the defect-hydrogel interface (Manuscript):

"To explore the binding force between A-C bioink and tissue surface, A30/5-C30/20 and A30/5-C300/20 were poured (about 2 mm height) and photocrosslinked between two pieces of fresh pig tendon (cross-section was about 3 cm × 1cm) and two pieces of fresh pig ribs (cross-section was about ϕ 1.6 cm), respectively. The binding stress was tested by the method of stretching the tissue-hydrogel-tissue structure by clipping two pieces of tissue in the opposite direction at 1 mm/min. The binding stress between A30/5-C30/20 and pig tendon could reach above 4000 Pa (Fig.4h) and the one between A30/5-C300/20 and pig rib could reach nearly 6000 Pa (Fig. 4j). Two steps occurred on A30/5-C300/20 and rib curve because A30/5-C300/20 was more suitable for compressing and would form crack during the stretching test. All the results indicated that A-C bioink would meet the requirement of strong binding force."

3. The figures including Figure 3 and Figure 5 are too small and the error bar in Figure 7 is blurred.

The size of **Fig.3** and **Fig.5** has been tuned, respectively. To keep the integrity of the results about the same part, the arrangement of the data figures was not changed. Instead, the words and the data figures were enlarged at their original locations.

The error bar in Fig.7 has been tuned.

4. In Emergency bag for portability, the freezing tubes could be directly heated using the body temperature. But what way could avoid the cold injury on the skin?

Heating freezing tubes on the skin can definitely cause cold injury. However, this heating method is just designed in very urgent cases, in which there are no portable battery and USB heater and an organ is terribly damaged. In normal situation, the freezing tubes are designed to be heated with USB heater, which has been described in the previous manuscript.

The previous related content is shown below:

Emergency bag for portability (Manuscript):

"In an accident, the freezing tubes were removed from the container and thawed with a tiny heating pad at 37 °C powered by a portable USB battery (Fig. 2f). For very urgent accidents without heating devices, they could be directly heated using the body temperature. Certainly, the cold injury on the patient's skin should be noticed....."

5. In addition to 405 nm blue flashlight, is it possible to consider using other light sources that are safer for this printing, such as green light and red light.

As discussed in the Answer 1, blue light has been verified to be very safe in either published contributions or current clinical treatment. Other lights can also be used in photocrosslinking printing process. However, different kinds of photoinitiators should be chosen to match the light wavelength. For example, for red or infrared light, some special photoinitiators need to be added into the GelMA precursor, such as benzylidene cycloketone-based two-photon initiator, P2CK (*Brigo, Laura, et al. Acta biomaterialia 55 (2017): 373-384.*). In further update of A-C bioink in future, more suitable and safer photocrosslinking strategy could be optimized with the development of bioprinting and biomaterial.

6. In Figure 4b-c, the scale bar should be added.

The scale bar (or glass side length) has been added.

Reviewer #3 (Remarks to the Author):

In this study, the authors propose in situ bioprinting with a bioink composed of GelMA microgels and a GelMA precursor solution. The authors show that by reducing the volume ratio of the precursor solution to the microgels as much as possible, the bioink could maintain its shape after being ejected from a nozzle, as observed in the case of the Brigham fluid. In addition, the authors show that the bioink strongly bonds onto a recipient tissue through reactions with the photo-generated aldehyde groups. The authors further demonstrate that BMSC-encapsulated bioink is effective for bone regeneration in a rat cranial defect model. Although the manuscript includes vital data, certain results and descriptions are unclear, and a few of the interpretations are debatable. Furthermore, the conclusions might be partially unsupported or insufficiently proven by the results.

Thanks for your valuable comments. We noticed the problems you proposed to our contribution. Because these questions were not provided with numbers, we have divided the previous comments into several questions and here are the answers to these problems below:

1. The critical issue in this study is that no data is provided on the thickness of the tissue that can be treated with their approach. As there are no vasculatures or microchannels in the printed constructs, the supply of oxygen and nutrients to the embedded cells in thicker tissue grafts can cause significant complications. These issues become further apparent while printing tissues with a higher cell density similar to the in vivo tissues, although the present study used microgels with a low cell density (Fig. 6c) for the treatment of the thin tissues, such as defects of rat cranial bones.

In terms of the vascularization in the hydrogel structure, we have described the further development of A-C bioink about vascularization in future in **Outlook** in previous manuscript and our initial exploration result in **Supplementary Note 8** and **Supplementary Fig. 12** in the previous supplementary information. In these parts, we co-cultured the A30/5 (GelMA microgels) loaded with human umbilical vein endothelial cells (HUVECs) and GelMA precursor solution loaded with breast tumor cells (MDA-MB-231s), which could release vascular endothelial growth factor (VEGF) to induce the vascularization of HUVECs. Surprisingly, the HUVECs in A30/5 showed obvious 3D sprout in the A-C structure. Therefore, by further modification, A-C bioink is potential to realize the vessel network establishment in the integrated induction environment in vivo in future.

The previous related content is shown below:

Outlook (Manuscript):

"Furthermore, in exploration, we found A component encapsulating endothelial cells could be vascularized by C component encapsulating tumor cells secreting vascular endothelial growth factor (VEGF)^{36,37} (Supplementary Note 8), which would probably

realize in-situ vascularization on organ defects."

Supplementary Note 8 (Supplementary Information):

"As discussed in the DISCUSSION, more complex therapy situations could happen, in which the vascularization in in-situ bioprinting could be required. To examine the vascularization capability in A-C bioink system, HUVECs-laden A components were electrosprayed and mixed with MDA-MB-231s-laden C component. Because of the VEGF releasing ability of MDA-MB-231s, HUVECs would tend to rapid grow and gradually form 3D sprout. 3D sprout from A component formed after 5-day co-culturing, which demonstrate the potential of vascularization capability of A-C bioink for further complicated therapy cases (Supplementary Fig. 12)."

Supplementary Fig. 12 (Supplementary Information):

In terms of the nutrition supplying problem in the hydrogel structure, the reviewer brought forward that nutrition supplying would be restricted in the A-C structures with larger size and higher cell density. In this work, the thickness of the in-situ bioprinted structure was about 1.5 mm. Actually, either in in-situ bioprinting or in in-vitro bioprinting, it is currently regarded as a common problem and a researching topic in 3D bioprinting field.

Fortunately, researchers in our lab or other labs have proposed a series of effective way to solve this problem. In our lab, Shao et al has published related papers to solve this problem by mixing some sacrificed gelatin microgels in the GelMA bioink. In the 37°C-culture environment, the sacrificed gelatin microgels would melt and form bigger pores in the photocrosslinked GelMA structure as nutrition channels. (Shao, L. et al.

Sacrificial microgel-laden bioink-enabled 3D bioprinting of mesoscale pore networks. *Bio-Design and Manufacturing* 3, 30-39 (2020). The sketch is shown below:

(A) Multilevel scale pore and respective functions in tissue engineering

(B) 3D bioprinting

Currently: dense hydrogel networks for structural support

Desirable: mesoscale pore networks

Limiting biological function

Effective oxygen, nutrient, and waste diffusion

Moreover, Shrike Yu Zhang from Harvard University also published corresponding contribution to solve this problem. The designed bioink contains two immiscible aqueous phases: a cell/GelMA mixture and poly(ethylene oxide) (PEO), which is photocrosslinked from the GelMA phase to fabricate a predesigned cell-laden hydrogel construct, while the pores as nutrition channels are formed by subsequently removing the PEO phase. (Ying, G.L. et al. *Aqueous Two-Phase Emulsion Bioink-Enabled 3D Bioprinting of Porous Hydrogels*. *Adv Mater* 30, e1805460 (2018)). The sketch is shown below:

In future, the nutrition supplying problem would be solved by more and more ways and A-C bioink would be further developed with continuously appearing method. To make it clearer, related description has been added in **Outlook (Manuscript)** as below:

Outlook (Manuscript):

"Moreover, nutrition/gas supplying problem in large-scale hydrogel structure is a common problem in either in-situ bioprinting or in-vitro bioprinting. Fortunately, researchers in our lab³⁸ or other labs³⁹ have proposed a series of effective way to solve this problem. Therefore, in future, next generation A-C bioink could be designed as the one with sacrificed component or phase separation component to form more nutrition network in the A-C structure."

2. The emergency bag is shown in Fig. 2; however, the volume of the bioink seems to be small, and it is unclear where and how large of a tissue defect can be treated using that.

Then section of **Emergency bag for portability** and the displayed picture **Fig.2e-2f** introduced the design and usage of emergency bag packing A-C bioink and auxiliary tools. However, the displayed image is just a showing example of the using of emergency bag, rather than the exact volumes/quantities in reality.

Fig.2e-2f:

Actually, the total volume of bioink is determined by the volumes and quantities of the applied freezing tubes. To take one of the most popular biomedical brand Corning as example, its freezing tubes own a series of volume specification (1.2 mL, 2.0 mL, 4.0 mL, 5.0 mL). At A-C bioink production end, the prepared A-C bioink can be loaded with different types of freezing tubes according to the production requirements. Moreover, at the rescuing end, in terms of the quantities of freezing tubes, rescuers can definitely take as many freezing tubes loading A-C bioink as possible according to the patient injury situation. Therefore, with the further development and mass production of A-C bioink, this novel bioink system would be feasible for the in-situ bioprinting of tissue defect as large as possible.

To make this part much clearer, related description has been added as below:

Emergency bag for portability (Manuscript):

"In practice, the total volume of bioink is determined by the volumes and quantities of the applied freezing tubes. At A-C bioink production end, the prepared A-C bioink can be loaded with different types of freezing tubes according to the production requirements. Moreover, at the rescuing end, in terms of the quantities of freezing tubes, rescuers can definitely take as many freezing tubes loading A-C bioink as possible according to the patient injury situation. Therefore, with the further development and mass production of A-C bioink, this novel bioink system would be feasible for the in-situ bioprinting of tissue defect as large as possible."

3. In addition, the authors state in the entire manuscript including the title that the key point of this study is in situ 3-D bioprinting. However, there is no data showing that bioprinting with their bioink is beneficial.

Actually, rather than "NO DATA", the WHOLE manuscript is talking about in-situ 3D bioprinting and the proposed A-C bioink is beneficial for it.

In Introduction, we introduced the superiority of in-situ bioprinting compared to in-vitro bioprinting and described the reason why in-situ bioprinting has been restricted for a long time. Then, in terms of the bioink aspect, we proposed a novel and effective bioink system, namely A-C bioink, to solve the problems of traditional bioink in in-situ bioprinting.

The previous related content is shown below:

Introduction (Manuscript):

".....bone treatments. Compared to organ implantation based on in-vitro 3D bioprinting, it has more advantages for its in-situ deposition feature (Supplementary Note 1)."

"However, in-situ bioprinting is rudimentary and has been restricted in clinical applications. Besides the lack of reliable in-situ bioprinters, one of the main reasons is that there is less suitable bioink meeting its special requirements....."

"Here, we will develop the bioconcrete bioink (A-C bioink) for in-situ bioprinting, the name of which comes from concrete for construction and its abbreviation comes from the two main components: aggregate (A) and cement (C) (Fig. 1, Supplementary Video. 1 and and Supplementary Video. 2)....."

After that, to show the feasibility and effectiveness of A-C bioink, we fully examined all aspects of it that could limit the treatment effect of in-situ bioprinting, including the portability, rheological properties, in-situ printability, complex mechanical properties, binding force with injured tissue, in-situ tissue regeneration induction ability, etc. All the results showed that the proposed A-C bioink is very suitable and potential to be applied in clinical in-situ bioprinting treatment in future.

The previous related content is shown below:

Emergency bag for portability (Manuscript):

"An emergency bag was designed to meet the storage and portability requirements (Fig. 2e). A component loading cells....."

Rheological robustness and mechanism (Manuscript):

"A-C bioink should own high rheological robustness in different temperature conditions to adapt to different accident situations of in-situ treatment. A30/5-C30/20....."

Printability in simulated in-situ bioprinting (Manuscript):

".....However, A-C bioink, thanks to its great rheological robustness, could form uniform filaments at the three temperatures..... The deposited A-C bioink could perfectly maintain the 3D structure while C bioink gradually melted and mixed with the "blood". Thus, A-C bioink would show great adaptation to the complex conditions around wounds. Moreover, A-C bioink was successfully printed with a commercial 3D bioprinter to establish a 3D cube with 12 layers and 7.5 mm height in the roughly controlled environment temperature (30 °C) and on the "wound" receiving basement."

Binding effect on the defect-hydrogel interface (Manuscript):

"In in-vitro bioprinting, 3D structures are crosslinked on the deposition platform, resulting in the lack of strong binding force with tissues after transplanting. During therapy, the shifting of the transplanted structures can be invalid or dangerous for patients. By contrast, in-situ deposited A-C bioink would form a strong binding force on the defect/hydrogel interface (Fig. 4d). This is because C component can display fluidity after contacting body temperature and easily infiltrate the defect vacancy, increasing the attaching sites on the interface and the internal friction (Fig. 4e)....."

Mechanical properties of A-C composite structure (Manuscript):

"Compared to the traditional method for establishing composite structures, namely, printing strengthen scaffolds followed by casting soft hydrogel, the method based on A-C bioink has obvious superiority..... "

Bone regeneration in cranial defects (Manuscript):

".....Histological observations at higher magnifications confirmed that the neo-formed bone with typical structure in BMSC-loaded group showed many more regions of new mature bone formation than the other groups, indicating BMSC-loaded A-C bioink could promote endogenous bone formation in a critical-size rat cranial defect model."

In-situ cranial repair of different defect morphologies (Manuscript):

"The actual clinical cases of organ defect are caused by all kinds of accidents, such as fires, traffic accidents, and military injuries. Thus, the 3D morphologies and sizes of the organ defects are very different. To examine the in situ bioprinting capability of A-

C bioink in a clinical setting, four rat “patients” with cranial defects with approximately rectangular, square, trapezoid, and triangular shapes (1.5 mm height) were created with a dental trephine (Fig. 8a, 8c).....”

4. The authors conducted in situ bioprinting of the bioink onto four different cranial bone defects. In the experiments, the bone generation should have been compared to a control experiment in which the bioink is manually loaded with a pipette to demonstrate that the bioprinting of their bioink is beneficial.

The bone generation effect is extremely important to verify the feasibility of an in-situ treatment bioink, so that we clearly examined the in-vivo bone generation effect with BMSC-loaded A-C bioink at different time points, compared to BMSC-unloaded A-C bioink and blank group. The results have verified the good treatment effect. In further in-situ bioprinting practice, A-C bioink also showed obvious potential in in-situ treatment and bone regeneration function.

The previous related content is shown below:

Bone regeneration in cranial defects (Manuscript):

“.....At 2nd week, Micro-computed tomography revealed new bone formation in BMSC-loaded A-C group (Fig. 7c). However, no obvious new bone was formed in blank group, and BMSC-unloaded group showed a very limited amount of bone regeneration. It was probably because hydrogels acted as a scaffold for relevant cells on the original defect location and provided more growing space. Furthermore, BMSC-loaded A-C group exhibited better bone regeneration efficacy with a higher BV/TV (Fig. 7b). At 4th week, the bone almost completely bridged the injured site in BMSC-loaded A-C group, and BMSC-unloaded group also exhibited more new bone tissue growth.....”

In-situ cranial repair of different defect morphologies

“ After 6 weeks of implantation, micro-computed tomography revealed that new bone was formed from the edge toward the center of the defects in all “patients” (Fig. 8d, 8e), which verified the high feasibility of A-C bioink in in-situ bioprinting therapy.”

However, we don’t think it’s necessary to take the effect difference of A-C bioink by “machine” operation from “hand” operation as key data. Actually, whichever way the securers use, for A-C bioink itself, both of the two ways are just for the “bioink movement”, rather than the “treatment effect” which is related to the biological component, though there would be different printing position accuracy. Besides, in terms of structural component, human arm is similar to the robotic arm, thus the movement way won’t definitely influence the treatment effect.

We summarized the in-vivo experiment results at the start of this project below, when we extruded and deposited A-C bioink on the organ defect with hands or robot. Actually, no obvious difference could be found in the aspect of the bone generation effect. (scale

bar=500 μm)

5. Furthermore, their approach is to use previously photocrosslinked microgels, and therefore the performance should be compared with the typical method of bioprinting during UV irradiation.

This paper is focused on the novel microgel-based BIOINK design for in-situ bioprinting. In the previous manuscript, we have compared the superiority of A-C BIOINK to TRADITIONAL BIOINK for IN-SITU bioprinting, including the better rheological robustness, printability, composite mechanical properties, etc. Actually, whether we use previously photocrosslinked microgels or not, either “this method” or “typical method” in the reviewer's question is EXTRUSION BIOPRINTING method with light irradiation substantially. Thus, our previous manuscript has done this part of work.

The previous related content is shown below:

Rheological robustness and mechanism (Manuscript):

"A-C bioink should own high rheological robustness in different temperature conditions to adapt to different accident situations of in-situ treatment....."

.....The thixotropy results by adding periodically varied oscillation amplitudes (200% and 1%) indicating the state transfer of A-C bioink was rapid and obvious, confirming its rapid self-healing speed. (Fig. 3g)

Thermo-sensitive bioinks need time to achieve stable state under certain temperature. Besides, for a certain temperature, the sol-gel state at certain time would be affected by the previous state and totally different during the temperature-increasing/decreasing process..... Furthermore, for such a wide temperature changing range, the viscosity of C bioink stretched 4–5 magnitudes, whereas that of A-C bioink maintained inside 1 magnitude because of the dominant role of microgels....."

Printability in simulated in-situ bioprinting (Manuscript):

"A simulated in-situ bioprinting scene was established to evaluate the printability of A-C bioink from the extruding and deposition states..... At 37 °C, C bioink was in an excessive solization state and form droplets. At 4 °C, C bioink was in an excessive gelation state and had become hydrogel bulk in the syringe, intermittently forming fragment. C bioink showed good printability only at 24 °C and formed uniform filaments. However, A-C bioink, thanks to its great rheological robustness, could form uniform filaments at the three temperatures. For deposition state, the environment temperature was set as 24 °C to achieve the best extruding state. To simulate the patients' wounds, the receiving platform was set as 37 °C (body temperature) and some food coloring solution (blood) was daubed (Fig. 4b). The deposited A-C bioink could perfectly maintain the 3D structure while C bioink gradually melted and mixed with the "blood". Thus, A-C bioink would show great adaptation to the complex conditions around wounds....."

Fig.3c (Manuscript):

Fig.3h (Manuscript):

Fig.4a-c (Manuscript):

Furthermore, microgel-based extrusion printing owns substantial priority to tradition bioink with only hydrogel precursor solution. This aspect has been discussed in **Introduction** of the previous manuscript. Corresponding published contributions have also been referred.

The previous related content is shown below:

Introduction (Manuscript):

".....Recently, besides independent function unit, in the review on microgels published in *Nature Reviews* by Burdick et al in 2020, the wide application of the “secondary bioprinting” of microgels as a bioink component in the future has been predicted. In latest work of Alge et al published in *Science Advances* in 2021, an in-depth investigation on the microgels dissipation process during printing was presented. Wang et al injected alginate microgels to repair rat organ defect. Burdick et al. extruded gathered microgels to establish specific 3D structures. All of the research benefited from not only the excellent biocompatibility of microgels but also their unique rheological properties similar to Bingham fluid, which displays as elastomer below certain stress but flows as Newton fluid once the stress was further increased. Therefore, microgel-based bioink have the potential to be further designed as a bran-new clinical in-situ bioprinting bioink to adapt the complicated requirements."

References (Manuscript):

14. Daly, A.C., Riley, L., Segura, T. & Burdick, J.A. Hydrogel microparticles for biomedical applications. *Nature Reviews Materials* 5, 20-43 (2019).
15. Agrawal, G. & Agrawal, R. Functional Microgels: Recent Advances in Their Biomedical Applications. *Small* 14, e1801724 (2018).
16. Chen, J. et al. 3D bioprinted multiscale composite scaffolds based on gelatin methacryloyl (GelMA)/chitosan microspheres as a modular bioink for enhancing 3D neurite outgrowth and elongation. *J Colloid Interface Sci* 574, 162-173 (2020).
17. Ouyang, L., Armstrong, J.P.K., Salmeron-Sanchez, M. & Stevens, M.M. Assembling Living Building Blocks to Engineer Complex Tissues. *Advanced Functional Materials* 30 (2020).

18. Song, K., Compaan, A.M., Chai, W. & Huang, Y. Injectable Gelatin Microgel-Based Composite Ink for 3D Bioprinting in Air. *ACS Appl Mater Interfaces* 12, 22453-22466 (2020).
19. Xin, S. et al. Generalizing hydrogel microparticles into a new class of bioinks for extrusion bioprinting. *Science Advances* 7, eabk3087.
20. Highley, C.B., Song, K.H., Daly, A.C. & Burdick, J.A. Jammed Microgel Inks for 3D Printing Applications. *Adv Sci (Weinh)* 6, 1801076 (2019).
21. Zhang, H. et al. Direct 3D Printed Biomimetic Scaffolds Based on Hydrogel Microparticles for Cell Spheroid Growth. *Advanced Functional Materials* 30, 1910573 (2020).
22. Ansley, R.W. & Smith, T.N. Motion of spherical particles in a Bingham plastic. *AIChE Journal* 13, 1193-1196 (1967).
23. Beverly, C.R. & Tanner, R.I. Numerical analysis of three-dimensional Bingham plastic flow. *Journal of Non-Newtonian Fluid Mechanics* 42, 85-115 (1992).
24. Yang, L. & Du, K. A comprehensive review on the natural, forced, and mixed convection of non-Newtonian fluids (nanofluids) inside different cavities. *Journal of Thermal Analysis and Calorimetry*, 1-22 (2019).

6. In sentence number 116, the authors state that “The volume proportion of A component was analyzed red/green areas as $73.215\% \pm 2.312\%$ (Supplementary Note 2), which was similar to the atomic space utilization in hexagonal closest packed (HCP) in crystal chemistry (74.05%) (Fig. 2d)” However, Fig. 2c shows relatively large areas of red fluorescence and black areas with no fluorescence, which appears to contradict this statement.

In previous manuscript, the black area in the A-C structure owned lower fluorescence signal because of the focus plane problem. To show it clearer, we have exchanged it into another image of another layer and showed it in **Fig.2** as below:

7. The authors emphasize the advantages of in situ bioprinting however concerns or limitations are not mentioned. For example, in situ bioprinting has the disadvantage of irradiating the wound with UV light and the photoinitiator cannot be removed and enters the body.

As the reviewer brought forward, the questions of UV irradiation and the residence of photoinitiator in patient body are definitely key aspects which need to be concerned. Actually, these questions are similar to question 1 because they are all common questions in the field of bioprinting, rather than only IN-SITU bioprinting. Thus, it is not the point or researching topic need to be mainly discussed in this paper. Fortunately, lots of researchers have been researching the damage of UV light and photoinitiator to cells or tissue during bioprinting and the biocompatibility of the combination of GelMA hydrogel, photoinitiator LAP and 405 nm blue light has been verified to be safe and widely applied in lots of published contributions (*ACS Appl. Mater. Interfaces* 2018, 10, 8, 6849–6857; Zhou, Feifei, et al. *Biomaterials* 258 (2020): 120287.; Pan, Jie, et al. *Journal of Materials Science: Materials in Medicine* 31.1 (2020): 1-12.; Wang, Guanhuier, et al. *Gels* 7.4 (2021): 247.). Furthermore, rather than UV light, we used 405 nm blue light in our work, which has been being widely applied in current clinical scenes, such as tooth photocrosslinking, blue light treatment of neonatal jaundice, etc. Therefore, the damage to the laden cells caused by 405 nm blue light can be ignored. For residence of photoinitiator in body, considering the fast development of bioprinting technology either in vitro or in situ, we believe that corresponding clinical authorities would examine the influence of residual photoinitiator according to standard testing and qualification process in future.

Related description has been added as below:

Emergency bag for portability (Manuscript):

“, which has been verified to be safe and widely used in current clinical scenes, such as tooth photocuring, blue light treatment of neonatal jaundice, etc.”

REVIEWERS' COMMENTS

Reviewer #1 (Remarks to the Author):

The author answered all the questions well. The article should be accepted

Reviewer #3 (Remarks to the Author):

The reviewer considers that the authors now produced a more balanced and better account of the work and that the revised manuscript is acceptable for publication.